

# Advanced methods for missing values imputation based on similarity learning

Khaled M. Fouad[1,2,*], Mahmoud M. Ismail[1,*], Ahmad Taher Azar[1,3] and Mona M. Arafa[1]

[1] Faculty of Computers and Artificial Intelligence, Benha University, Benha, Qaliobia, Egypt
[2] Faculty of Information Technology and Computer Science, Nile University, El Shikh Zaid, Giza, Egypt
[3] College of Computer & Information Sciences, Prince Sultan University, Riyadh, Saudi Arabia
[*] These authors contributed equally to this work.

Corresponding author
Mahmoud M. Ismail,
mahmoud.ismael@fci.bu.edu.eg

## ABSTRACT

The real-world data analysis and processing using data mining techniques often are facing observations that contain missing values. The main challenge of mining datasets is the existence of missing values. The missing values in a dataset should be imputed using the imputation method to improve the data mining methods' accuracy and performance. There are existing techniques that use k-nearest neighbors algorithm for imputing the missing values but determining the appropriate k value can be a challenging task. There are other existing imputation techniques that are based on hard clustering algorithms. When records are not well-separated, as in the case of missing data, hard clustering provides a poor description tool in many cases. In general, the imputation depending on similar records is more accurate than the imputation depending on the entire dataset's records. Improving the similarity among records can result in improving the imputation performance. This paper proposes two numerical missing data imputation methods. A hybrid missing data imputation method is initially proposed, called KI, that incorporates k-nearest neighbors and iterative imputation algorithms. The best set of nearest neighbors for each missing record is discovered through the records similarity by using the k-nearest neighbors algorithm (kNN). To improve the similarity, a suitable k value is estimated automatically for the kNN. The iterative imputation method is then used to impute the missing values of the incomplete records by using the global correlation structure among the selected records. An enhanced hybrid missing data imputation method is then proposed, called FCKI, which is an extension of KI. It integrates fuzzy c-means, k-nearest neighbors, and iterative imputation algorithms to impute the missing data in a dataset. The fuzzy c-means algorithm is selected because the records can belong to multiple clusters at the same time. This can lead to further improvement for similarity. FCKI searches a cluster, instead of the whole dataset, to find the best k-nearest neighbors. It applies two levels of similarity to achieve a higher imputation accuracy. The performance of the proposed imputation techniques is assessed by using fifteen datasets with variant missing ratios for three types of missing data; MCAR, MAR, MNAR. These different missing data types are generated in this work. The datasets with different sizes are used in this paper to validate the model. Therefore, proposed imputation techniques are compared with other missing data imputation methods by means of three measures; the root mean square error (RMSE), the normalized root mean square error (NRMSE), and the mean absolute error (MAE). The results show that

the proposed methods achieve better imputation accuracy and require significantly less time than other missing data imputation methods.

## INTRODUCTION

Organizations today depend heavily on data gathering, storing, and processing for different decision-making processes (*Müller, Naumann & Freytag, 2003*; *Razavi-Far, Zio & Palade, 2014*). Data is gathered in several different ways, such as documents, surveys, sensors, interviews, and observations (*Chapman & Speers, 2005*; *Rahman & Islam, 2011*). According to various causes, such as human error, confusion, misunderstanding, equipment faults, measurement error, noise generation during transformation, and non-response, data may be lost or interrupted (*Rahman & Islam, 2013a*). If the data gathered is not completed, issues may occur in the decision-making process. An incomplete dataset may also affect data mining models' performance, resulting in a lack of computing process efficiency and an invalid and inefficient outcome due to dataset gaps (*Salleh & Samat, 2017*). The main challenge of mining datasets is the existence of missing values (*Poolsawad et al., 2012*). Extracting valuable information and knowledge from incomplete datasets is difficult (*Houari et al., 2014*). The preprocessing approach would play a significant role in the process of data mining (*Fouad, El Shishtawy & Altae, 2018*). Therefore, it is essential to clean the dataset to ensure the high-quality mining (*Han & Kamber, 2013*; *Sree Dhevi, 2014*).

The dataset is imputed from missing values using the imputation method, which is one of the effective approaches to solve that issue and improve the accuracy and performance of the data mining techniques. The accurate estimation of missing values plays a vital role in ensuring a high level of data quality in many areas, such as healthcare (*Azimi et al., 2019*) and traffic monitoring (*Li et al., 2020*). Identifying patterns of missing data is a crucial factor when developing strategies for tackling incomplete data. In particular, the type of missing data can significantly affect the accuracy of data mining techniques. The missing data can be divided into three categories (*Soley-bori, 2013*; *Salgado et al., 2016*; *Garciarena & Santana, 2017*). The first category is missing completely at random (MCAR). In MCAR, when missing in the dataset occurs entirely at random, no specific pattern can be determined. For example, some patients may have lost laboratory values due to incorrect handling of several lab results. The second category is missing at random (MAR). In MAR, a specific pattern can be determined. The possibility that a particular variable's value is missed or not for any observation depends on the values of other variables; therefore, a common factor can be found in all observations that have missing values. For example, a depression-examination registry may encounter data representing MAR if males are less likely to complete a questionnaire about the severity of depression than females. Therefore, the likelihood of completion of the

questionnaire is related to their gender (that is fully observed). The third category is missing not at random (MNAR). In MNAR, missing data is neither MCAR nor MAR. In this case, the data that cause others to be missing are unobserved. Following the previous example, the depression-examination registry may encounter data representing MNAR if people with extreme depression are more likely to refuse to complete the questionnaire about the severity of depression.

Varieties of methods that impute the missing values have been presented (*Zhang et al., 2006*). In general, imputation performance depends heavily on choosing an acceptable imputation method to impute missing data (*Zhang et al., 2006*). The performance of each imputation method can be varied dependent on the types of missing data and datasets. In the imputation of the missing data, the current approaches usually use the similarities of missing rows with the other rows in the dataset and the correlations of the features (*Sefidian & Daneshpour, 2020*). Therefore, the missing data imputation methods can be partitioned into two main categories, which are global missing data imputation methods and local missing data imputation methods (*Cheng, Law & Siu, 2012*; *Feng et al., 2015*). The global missing data imputation includes the strategies that use the whole dataset's global correlation structure to impute missing values found in the dataset. Several current imputation methods, such as iterative imputation (*Little & Rubin, 2002*; *Van Buuren & Groothuis-Oudshoorn, 2011*; *Pedregosa et al., 2011*) and expectation-maximization imputation (EMI) (*Schneider, 2001*; *Junninen et al., 2004*), are considered in this category that was described in (*Rahman & Islam, 2013b*). The iterative imputation is a multivariate imputation that uses the whole set of available features to predict missing values. It is a sophisticated method that models each feature, which has missing values as a function of other features in a round-robin fashion (*Little & Rubin, 2002*; *Van Buuren & Groothuis-Oudshoorn, 2011*; *Pedregosa et al., 2011*). The local missing data imputation includes the strategies that use only the records similar to the missing record to impute missing values such as the k-nearest neighbor imputation (kNNI) (*Batista & Monard, 2003*). kNNI is an effective method to impute missing values. However, it is expensive for a large dataset because it is required to search within the entire dataset to find the most similar records. Moreover, k value is determined by the user so determining the appropriate k value can be a challenging task (*Batista & Monard, 2003*; *Rahman & Islam, 2013b*; *Liu et al., 2015*). Recently, a number of methods based on k-means clustering algorithm have been proposed to solve the problem of missing data imputation (*Patil, Joshi & Toshniwal, 2010*; *Jiang & Yang, 2015*). The basic idea behind these techniques is to estimate a missing value in a record based on the cluster information in which the missing record is located. But in the hard clustering, such as the k-means algorithm, a record belongs to only one cluster. When records are not well-separated, as in the case of missing data, hard clustering provides a poor description tool in many cases. Besides, if the initial points are not chosen properly, the k-means algorithm may become stuck in a local minimum state (*Sefidian & Daneshpour, 2019*). There are group of missing data imputation techniques that make utilization of a decision tree for the horizontal partitioning such as DMI (*Rahman & Islam, 2011*) and KDMI (*Rahman & Islam, 2013b*). This is computationally expensive (*Razavi-Far et al., 2020*). Moreover, a decision tree

algorithm may generate some heterogeneous leaves, where the records in each heterogeneous leaf are not always very similar. This may lead to poor imputation accuracy (*Rahman & Islam, 2014*). In general, the imputation depending on similar records is more accurate than the imputation depending on the entire dataset's records (*Young, Weckman & Holland, 2011*). Improving the similarity among records can result in improving the imputation performance. Another issue of interest is finding a strategy that can impute the missing values in large-scale datasets (*Fouad & Elbably, 2020*). A missing data imputation method should be time-efficient, which means that it should not rely on the entire dataset for imputing missing records (*Razavi-Far et al., 2020*).

This paper initially proposes a hybrid missing data imputation method, called KI, that consolidates k-nearest neighbors and iterative imputation algorithms to impute the missing values in a dataset. The best set of nearest neighbors for each missing record is discovered based on the similarity of records by the kNN algorithm. To improve the similarity, a suitable k value is estimated automatically for the kNN without any user input. The iterative imputation method is then used to impute the missing values of the incomplete records by using the global correlation structure among the selected records. This technique applies only one level of similarity. It can improve missing data imputation accuracy. However, it is expensive for a large dataset because it is required to search within the entire dataset to find the most similar records for each missing record.

This paper then proposes another enhanced hybrid missing data imputation method, called FCKI, that is an extension of KI. It integrates fuzzy c-means, k-nearest neighbors, and iterative imputation algorithms for imputing the missing data for a dataset. This technique focuses on improving time efficiency for the proposed missing data imputation algorithm as well as missing data imputation accuracy. It performs fuzzy c-means clustering for the dataset to divide the dataset records into a c fuzzy clusters where the records in the same cluster are similar. Then, it imputes each cluster separately using KI algorithm through two phases. In the first phase, the best set of nearest neighbors for each missing record is discovered based on the similarity of records by the kNN algorithm. Phase two focuses on exploiting the iterative imputation method to impute the missing values of the incomplete records by using the global correlation structure among the selected records. FCKI differs from KI, in that FCKI applies two levels of similarity to achieve a higher imputation accuracy before imputing the missing values through the iterative imputation. For the first level of similarity, fuzzy c-means clustering is selected. The similarities of all records belonging to the cluster are higher than the similarities of all the whole dataset records. The fuzzy c-means algorithm is selected because the records can belong to multiple clusters at the same time. This can lead to further improvement for similarity. For the second level of similarity, kNN is selected. It finds the best k records that are the utmost similar to the missing record by using the Euclidean distance measure. FCKI, similar to KI finds the best k value for the kNN automatically. FCKI searches a cluster, instead of the whole dataset, to find the best k-nearest neighbors. The technique has the advantage of imputing missing values based on the similarity of the set of records instead of the whole dataset. Iterative imputation is then applied to discover the global correlation among the selected records similar to each other to impute missing values.

FCKI can improve time efficiency because It does not have many iterations for imputing missing values in the dataset. It also focuses on both the similarity of data records and the correlation among the features. Therefore, it can enhance imputation efficiency and effectiveness where the most efficient imputation method should impute incomplete dataset with the least amount of time, and the most effective imputation method should achieve the highest imputation accuracy. The effectiveness of the imputation can be checked using different imputation performance measures.

The scalability of the clustering algorithm is becoming increasingly important in today's data mining applications due to the growing size of datasets (*Ganti, Gehrke & Ramakrishnan, 1999*). It's natural for a single processor machine to be unable to store the entire dataset in main memory for processing, and frequent disk access results in a performance bottleneck. Scalable and high-performance solutions can now be easily accomplished by implementing parallel clustering algorithms, especially after the recent development of affordable parallel computing platforms. *Kwok et al. (2002)* proposed the parallel fuzzy c-means (PFCM) algorithm for clustering large datasets. parallel computers of the Single Program Multiple Data model type with the Message Passing Interface are used to run this algorithm (*Kwok et al., 2002*). The PFCM algorithm can be used for FCKI algorithm instead of traditional FCM for clustering large datasets.

It can be said that the proposed imputation methods, KI and FCKI, follow a hot deck approach. Hot deck imputation includes using observed values from a respondent (the donor) who is close to the non-respondent (the recipient) to fill in missing values for one or more variables for a non-respondent (*Andridge & Little, 2010*).

The proposed imputation methods, KI and FCKI, consider datasets with missing values in multiple numerical features. They are assessed by using fifteen known datasets with variant missing ratios ranging from 1% to 20% of total attribute values for each type of missing data; MCAR, MAR, MNAR. These different missing data types are generated in this work. The datasets with different sizes are used in this paper to validate the model. These sizes are gradually increased from small to large-scale. Small data is data in a volume and format that makes it accessible, informative, and actionable. Large-scale data can be associated with data that grows to a huge size over time. The proposed techniques are compared to different ten missing data imputation methods, which are mean imputation (*Ravi & Krishna, 2014*), kNNI (*Batista & Monard, 2003*; *Rahman & Islam, 2013b*; *Liu et al., 2015*), SoftImpute (*Mazumder, Hastie & Tibshirani, 2010*), SVDImpute (*Troyanskaya et al., 2001*), traditional iterative imputation (*Little & Rubin, 2002*; *Van Buuren & Groothuis-Oudshoorn, 2011*; *Pedregosa et al., 2011*), EMI (*Schneider, 2001*; *Junninen et al., 2004*), DMI (*Rahman & Islam, 2011*), KDMI (*Rahman & Islam, 2013b*), KEMI (*Razavi-Far et al., 2020*) and KEMI[+] (*Razavi-Far et al., 2020*). The evaluation of the proposed missing data imputation methods is performed by using three imputation performance measures. These measures are the root mean square error (RMSE), the normalized root mean square error (NRMSE), and the mean absolute error (MAE).

The structure of the paper is as follows. "Related Work" provides some related works on missing data imputation. "Algorithms Used to Generate Missing Data" gives a formal

presentation of the algorithms used to generate different missing data types. The proposed missing data imputation methods, KI and FCKI, are formally presented in "Proposed Missing Data Imputation Methods". "Results and Discussion" presents results and discussion and compares KI and FCKI with other missing data imputation methods. "Conclusions and Future Work" gives the conclusions and future work.

## RELATED WORK

Recently, the imputation of missing values has attracted more and more attention from researchers. There are two primary techniques are used to impute missing data. The first technique aims at ignoring records that contain missing values. Although it is simple, it is an inefficient method, particularly through high missing rate datasets. The second primary technique is estimating the missing values, so-called missing data imputation (*Bethlehem, 2009*). Several missing data imputation methods have been proposed, and they can show significant variations in terms of complexity and quality of the imputation. This section presents several missing data imputation methods.

Mean imputation is the most basic method used by previous researchers. It replaces the missing value with the mean of non-missing values for the attribute. If there are several missing values in an attribute, they all will be replaced by the same value because mean imputation generates only one imputed value. It does not preserve the correlation among the features. As most research studies are concerned with the relationships among features, mean imputation is not a good solution (*Ravi & Krishna, 2014*).

Hot deck imputation includes using observed values from a respondent (the donor) who is close to the non-respondent (the recipient) to fill in missing values for one or more variables for a non-respondent. The donor is chosen at random from a group of possible donors in some versions. These techniques are named random hot deck techniques. In other variants, a single donor is defined, and values are estimated from that case, typically the "nearest neighbor". Since no randomness is involved in the selection of the donor, these methods are known as deterministic hot deck methods (*Andridge & Little, 2010*). One of the most common deterministic hot deck imputation methods is sequential nearest neighbor hot deck imputation. Traditional hot deck imputation is another name for this form. The first step in this method is to define imputation classes using some auxiliary variables. Second, a single value, such as the class mean or any pre-specified value, is assigned as a starting point for each imputation class. The data file's records are then treated in sequence. If a record has a response for the target variable, that value takes the place of previously stored value for its imputation class. If a record has a missing value for a target attribute, the value currently stored for its imputation class is assigned to it. The major drawback of this method has is that it is more likely to leads to multiple use of donors, a feature which leads to a loss of precision in estimation (*Hu & Salvucci, 2001*).

k-nearest neighbor imputation (kNNI) is an effective method to impute missing values. It first identifies k-nearest neighbors, which are the most similar to the missing record among all records within the dataset by using the euclidean distance (k is determined by the user). kNNI uses the mean value of the feature, which has the missing value within the selected nearest neighbors. KNNI imputation accuracy is better than mean imputation

accuracy which calculates the mean from the entire dataset instead of the k-nearest neighbors of missing record within the dataset. kNNI is an efficient and unpretentious method. However, it is expensive for a large dataset because it is required to search within the entire dataset to find the most similar records. Moreover, determining the appropriate k value can be a challenging task (*Batista & Monard, 2003*; *Rahman & Islam, 2013b*; *Liu et al., 2015*).

The SVDimpute algorithm uses a low-rank SVD approximation to impute the missing values in an incomplete dataset. It firstly initializes all missing values to the column means. Until convergence, it computes a rank-k approximation to the completed matrix. It replaces the previously missing values with corresponding values from the rank-k approximation obtained in the previous step (*Troyanskaya et al., 2001*).

SoftImpute is used for solving the problem of matrix completion. It fits a low-rank matrix approximation to the matrix, which contains the missing values via nuclear-norm regularization. It makes use of soft-thresholded SVD to impute missing values (*Mazumder, Hastie & Tibshirani, 2010*).

Iterative imputation is a multivariate imputation that uses the whole set of available features to predict missing values. A sophisticated method that considers each feature that has missing values depends on other features, and this estimation is used for imputation. It does so in an iterated round-robin fashion: at each iteration, a feature with missing values is specified as output y, and the other features are handled as inputs x. A regressor is fit on (x, y) for known y. The regressor is then used to estimate the missing values of y. This is iteratively executed for each feature and is repeated for several imputation rounds. The final imputation round results are returned (*Little & Rubin, 2002*; *Van Buuren & Groothuis-Oudshoorn, 2011*; *Pedregosa et al., 2011*).

Expectation-Maximization Imputation (EMI) algorithm uses the mean and covariance matrix of the dataset for imputing the missing numerical values of the incomplete dataset. It firstly computes the derived matrix that includes the mean and the covariance values of the dataset, which contains missing values. It then imputes the missing values through the mean and covariance matrix (*Schneider, 2001*; *Junninen et al., 2004*). The main downside of this approach is that it uses information from the entire dataset to impute the missing value, so it is only appropriate for the datasets which have strong correlations between the attributes (*Deb & Liew, 2016*).

There are group of techniques that use a decision tree for imputing the missing data such as DMI (*Rahman & Islam, 2011*) and KDMI (*Rahman & Islam, 2013b*). DMI is an imputation technique that is based on the decision tree to impute the missing data. DMI consolidates the decision tree and the expectation-maximization algorithm (EM). Therefore, DMI split the dataset into two sub-datasets. The first sub dataset contains records with no missing values, and the second one contains records with missing values. It creates a collection of decision trees on the first sub dataset considering the features which have missing values in the second sub dataset as the class attributes. It assigns each missing record of the second sub dataset to the leaf where it exists for the tree that considers the attribute, which has a missing value for the record, as the class attribute. It lastly uses the EMI algorithm (*Schneider, 2001*; *Junninen et al., 2004*) for imputing the

missing values among the records within each leaf (*Rahman & Islam, 2011*). KDMI is an enhanced version of the DMI. It has two levels of partitioning for a dataset. The first phase is similar to the DMI algorithm, where it horizontally splits the data into a collection of portions generated by decision tree leaves. The second phase uses the kNN algorithm to identify a set of nearest neighbors, which are extremely similar to the missing record through all records that exist in the decision tree leaf where the missing record is found in. It eventually uses the EMI algorithm for imputing the missing values using the selected nearest neighbors (*Rahman & Islam, 2013b*). DMI and kDMI make utilization of a decision tree for the horizontal partitioning. This is computationally expensive. They build a decision tree for each feature having missing values in the dataset, considering this feature as a class attribute. If there are a large number of features having missing values, DMI and KDMI will build a large number of trees, even if the dataset contains a small number of records (*Razavi-Far et al., 2020*). A decision tree algorithm may generate some heterogeneous leaves, where the records in each heterogeneous leaf are not always very similar. This may lead to poor imputation accuracy (*Rahman & Islam, 2014*). DMI and KDMI divide the entire dataset into complete and incomplete records and then build trees from complete records only, so if the dataset does not contain any complete record, the algorithms will not be able to build any tree and will not be able to impute any incomplete records. Therefore, there must be a minimum number of complete records for the imputation process to be performed correctly. Another issue is that it is unclear how the imputation is handled if the missing record falls into more than one leaf, which may happen if a record has several missing values (*Deb & Liew, 2016*).

A number of methods based on k-means clustering algorithm have been proposed to solve the problem of missing data imputation (*Patil, Joshi & Toshniwal, 2010*; *Jiang & Yang, 2015*). *Patil, Joshi & Toshniwal (2010)* proposed an efficient missing value imputation method based on k-means clustering with weighted distance. They use the user-specified value k to divide the dataset into clusters. Then determine the complete neighbor which is similar to the missing instance. The missing value is then estimated by taking the average of the centroid value and the neighbor's centroidal distance. *Jiang & Yang (2015)* proposed an improved KNN based algorithm, called class-based k-clusters nearest neighbor imputation (CKNNI). The k-means cluster algorithm and traditional kNN algorithm are used for imputing missing data. First, CKNNI uses the k-means algorithm to cluster instances into clusters. Then, the nearest neighbor from the collection of centroids in clusters is selected using kNN algorithm, and with the ones from corresponding attributes in a chosen neighbor, missing values are imputed. The limitation of using hard clustering, such as the k-means algorithm, is that a record belongs to only one cluster. When records are not well-separated, as in the case of missing data, hard clustering provides a poor description tool in many cases. Besides, if the initial points are not chosen properly, the original k-means algorithm may become stuck in a local minimum state (*Sefidian & Daneshpour, 2019*).

*Razavi-Far et al. (2020)* proposed two methods for missing data imputation, named KEMI and KEMI[+]. KEMI uses the k-nearest neighbors algorithm and expectation-maximization imputation (EMI) algorithm to tackle missing values in an incomplete

dataset. It firstly uses the kNN algorithm to identify the best set of nearest neighbors, which are the most similar to the missing record among all records that don't contain any missing value. Then, EMI is used to impute the missing values using the selected nearest neighbors (*Razavi-Far et al., 2020*). KEMI$^+$ is an enhanced version of the KEMI. It is used to identify the collection of superior k values that leads to the minimum imputation error. It uses EMI to derive the missing values using a collection of top nearest neighbors. It eventually provides a collection of superior estimations into the dempster–shafer fusion to fuse these estimations and return final estimation (*Razavi-Far et al., 2020*). KEMI and KEMI$^+$ divide the entire dataset into complete and incomplete records and then impute the missing values using only complete records, so if the dataset does not contain any complete record, the algorithms will stop and will not be able to impute any incomplete record. Therefore, there must be a minimum number of complete records for the imputation process to be performed correctly. Moreover, KEMI and KEMI$^+$ are expensive because they search within all complete records of the dataset to find the most similar records for each missing record.

# ALGORITHMS USED TO GENERATE MISSING DATA

As mentioned earlier, missing data has various types. The original datasets that are used in this paper are not incomplete datasets. The specific types of missing values are generated in this section based on three described algorithms. These algorithms are used to generate missing values in a predefined ratio for datasets to simulate the various missing data types; MCAR, MAR, MNAR. These algorithms are derived from *Garciarena & Santana (2017)*. For each of the three types of missing data mentioned in this paper, one method has been implemented. For the experimentation part, we decided to add missing data with different missing ratios ranging from 1% to 20% of total attribute values. These are reasonable values for the amount of missing data in real datasets. These algorithms randomly generate missing data to achieve the delivered datasets that are nearly real. The dataset positions to be modified are selected randomly according to the specified conditions for each missing data type. It aims to generate instances of the dataset with variant missing values for each execution of the algorithm.

## MCAR

The algorithm used to generate MCAR is straightforward, where it first generates two random numbers and used them as coordinates in the dataset. Then, it changes the value indicated by these coordinates to "NaN". This process is iterated until the predefined missing ratio is reached. The pseudo-code is presented in Algorithm 1.

## MAR

The algorithm used to generate MAR is less straightforward than the previous one, where it needs to determine which attribute is the causative of the missing data. *Garciarena & Santana (2017)* presumes a single attribute as causative, but causative attributes of MAR can be multiple. Therefore, the algorithm first selects the causative attribute randomly and reproduces its values to a new vector (aux). The next step is to select

---

**Algorithm 1  MCAR generating algorithm.**

**Input:**

D: A complete dataset

MDR: Missing data ratio

**Output:** Incomplete dataset with MDR % missing ratio of total attribute values of D

**Definitions:**

N is the total number of records in D

M is the total number of attributes in D

X is the index of a random record

Y is the index of a random attribute

R is the current missing ratio of total attribute values of D

**Begin**

  **while** true **do**

    $X \leftarrow$ Random([0,N)) /* Generate random record index between 0 and N */

    $Y \leftarrow$ Random([0,M)) /* Generate random attribute index between 0 and M */

    **if** D[X,Y] != NaN **then**

      D[X,Y] = NaN

      $R \leftarrow$ GetCurrentMDRatio(D) /* Return the current missing ratio of total attribute values of D */

      **If** $R \geq$ MDR **then**

        **break**

      **end**

    **end**

  **end**

  Return incomplete dataset D

**end**

---

randomly the attributes that will lose their values (MDAttributes). These attributes are called dependent because causative attribute tends to cause their values to be lost. It then determines which records will lose their values for the dependent attributes (MDRecords) by selecting the minimum values for the causative attribute (aux) and making them incompetent by allocating them a great number. The process of choosing records is repeated until the predefined missing ratio is reached. Finally, it uses the nested loop to generate the missing data using the selected attributes and the selected records. The pseudo-code is presented in Algorithm 2.

## MNAR

The algorithm used to produce MNAR is quite similar to the one used for producing MAR. In MNAR, the causative attribute is unknown. Therefore, the records that will lose their values will be selected randomly. The pseudo-code is presented in Algorithm 3.

---

**Algorithm 2** MAR generating algorithm.

**Input:**

D: A complete dataset

MDR: Missing data ratio

NA: Attributes number losing their values

**Output:** Incomplete dataset with MDR % missing ratio of total attribute values of D

**Definitions:**

N is the total number of records in D

M is the total number of attributes in D

V is the total number of dataset values

Causative is the attribute that causes data loss in other attributes

MV is the total number of missing values

X is the index of a random record

Y is the index of a random attribute

R is the current missing ratio of total attribute values of D

MDAttributes is the list of attributes that lose their values

MDRecords is the list of records that lose their values

Aux is the vector of causative attribute values

MinIndex is the index of the minimum value in Aux

**Begin**

  V = N * M /* Calculate the total number of dataset values */

  Causative ← Random([0,M]) /* Generate random attribute index between 0 and M */

  MDAttributes = []

  **for** i = 1, . . . , NA **do**

    **do**

      Y ← Random([0,M]) /* Generate random attribute index between 0 and M */

    **while** Y in MDAttributes

    MDAttributes.Append(Y) /* Append Y to MDAttributes */

  **end**

  MDRecords = []

  MV = 0

  Aux = D[:,Causative] /* Return the vector of causative attribute values */

  **while** true **do**

    MinIndex ← FindMinIndex(Aux) /* Find the index of the minimum value in Aux */

    MDRecords.Append(MinIndex) /* Append MinIndex to MDRecords */

    Aux[MinIndex] = MaxInt /* MaxInt is the maximum integer that can be interpreted to a programming language, and it is utilized to deliver this index ineligible */

---

(Continued)

| Algorithm 2 (continued) |
| --- |

      MV += NA /* Missing values are incremented by the number of attributes which lose their values */

      R = (MV / V)*100 /* Calculate the current missing ratio of total attribute values of D */

      **If** R ≥ MDR **then**

          **break**

      **end**

   **end**

   **foreach** record i ∈ MDRecords **do**

      **foreach** attribute j ∈ MDAttributes **do**

       D[i,j] = NaN

      **end**

   **end**

   Return incomplete dataset D

**end**

# PROPOSED MISSING DATA IMPUTATION METHODS

This paper initially proposes a hybrid missing data imputation method, called KI. It consolidates k-nearest neighbors and iterative imputation algorithms to impute the missing values in a dataset. The general scheme for KI is visualized in Fig. 1. The best set of nearest neighbors for each missing record is discovered based on similarity of records using the kNN algorithm. The iterative imputation method is then used to impute the missing values of the incomplete records by using the global correlation structure among the selected records. This technique applies only one level of similarity. It can improve missing data imputation accuracy. However, it is expensive for a large dataset because it is required to search within the entire dataset to find the most similar records for each missing record.

The kNN algorithm is selected because it is commonly one of the utmost machine learning methods to search for local similarities. It finds the best k records that are the utmost similar to the missing record based on the Euclidean distance measure (*Santos et al., 2020*).

In KI, an appropriate k value is estimated automatically for the kNN. Initially, the algorithm creates a random missing value $r_{iz}$ for a record $R_i$ that includes an actual missing value $r_{ij}$. For each possible k value (ranging from 2 to N where N is the number of records), kNN finds the k-most similar records to $R_i$. The algorithm then imputes the missing value of $r_{iz}$ using the mean of the $z^{th}$ attribute over all records related to k-nearest neighbors of $R_i$. The algorithm then calculates the root mean squared error (RMSE) using the imputed value and the actual value of $r_{iz}$. Note that RMSE values for all sets of k-nearest neighbors of $R_i$ for the same $r_{iz}$ are computed. The best k value is extracted from the set of k-nearest neighbors that deliver the minimal value of RMSE (*Rahman & Islam, 2013b*).

**Algorithm 3 MNAR generating algorithm.**

**Input:**

D: A complete dataset

MDR: Missing data ratio

NA: Attributes number losing their values

**Output:** Incomplete dataset with MDR % missing ratio of total attribute values of D

**Definitions:**

N is the total number of records in D

M is the total number of attributes in D

V is the total number of dataset values

MV is the total number of missing values

X is the index of a random record

Y is the index of a random attribute

R is the current missing ratio of total attribute values of D

MDAttributes is the list of attributes that lose their values

MDRecords is the list of records that lose their values

**Begin**

  V = N * M /* Calculate the total number of dataset values */

  MDAttributes = []

  **for** i = 1, . . . , NA **do**

    **do**

      Y ← Random([0,M]) /* Generate random attribute index between 0 and M */

    **while** Y in MDAttributes

    MDAttributes.Append(Y) /* Append Y to MDAttributes */

  **end**

  MDRecords = []

  MV = 0

  **while** true **do**

    X ← Random([0,N]) /* Generate random record index between 0 and N */

    **if** X not in MDRecords **then**

      MDRecords.Append(X) /* Append X to MDRecords */

     MV += NA /* Missing values are incremented by the number of attributes which lose their values */

      R = (MV / V)*100 /* Calculate the current missing ratio of total attribute values of D */

      **If** R ≥ MDR **then**

        **break**

      **end**

    **end**

  **end**

  **foreach** record i ∈ MDRecords **do**

    **foreach** attribute j ∈ MDAttributes **do**

      D[i,j] = NaN

    **end**

  **end**

  Return incomplete dataset D

**end**

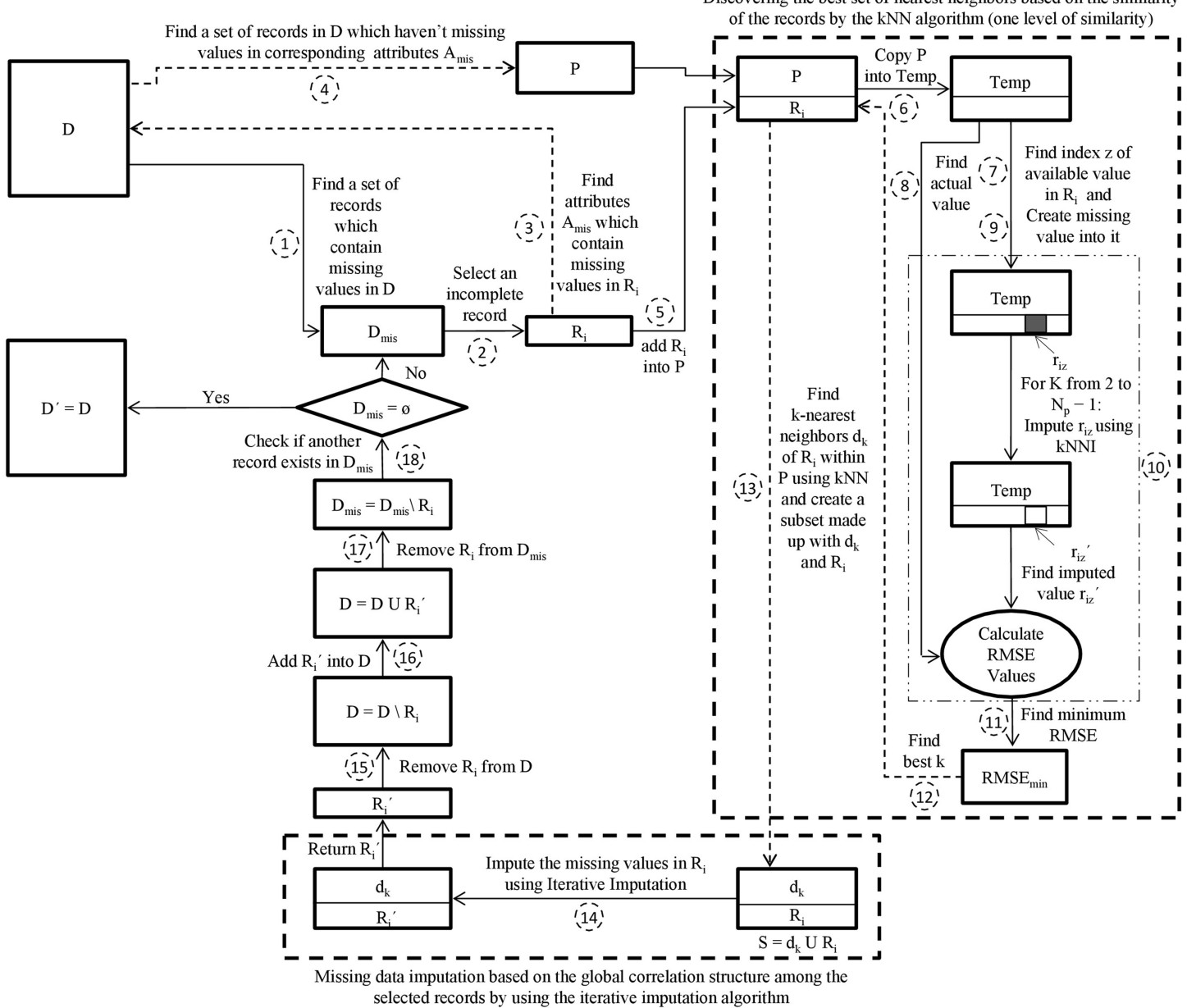

**Figure 1 The proposed KI scheme.** D is the incomplete dataset, D' is the imputed dataset of D, $D_{mis}$ is the incomplete subset of D, $R_i$ is the incomplete target record from $D_{mis}$, $R_i'$ is the imputed record of $R_i$, $A_{mis}$ is the set of attributes that contain missing values in $R_i$, P is the pool, Temp is a copy of the pool P, $r_{iz}'$ is the imputed value of $r_{iz}$, $N_p$ is the number of records in P, $d_k$ is k nearest neighbors of $R_i$, S is a subset made up of $d_k$ and $R_i$.

The pseudo-code of KI is presented in Algorithm 4. The definitions of notations used in Algorithm 4 are presented in Table 1. The main steps of the KI algorithm are explained in the following:

Step 1—Create a subset $D_{mis}$ that contains a set of records in D which have missing values.

Step 2—Select an incomplete record $R_i$ from $D_{mis}$.

Step 3—Find a set of attributes $A_{mis}$ which contain missing values in $R_i$.

Step 4—Create a pool P that contains the set of records in D which haven't missing values in corresponding attributes $A_{mis}$.

Step 5—Add the incomplete record $R_i$ into the pool $P = P \cup R_i$.

Step 6—Copy the pool P into Temp.

Step 7—Find a random index z of an attribute within $R_i$ for which the value of $r_{iz}$ is available.

Step 8—Preserve the value $r_{iz} \in R_i$ into a variable of actual value (AV).

Step 9—Create missing value into $r_{iz}$ within Temp.

Step 10—For each k value (ranging from 2 to $N_p - 1$ where $N_p$ represents the number of records in P), find the k-nearest neighbors $d_k$ of $R_i$ within Temp by using kNN algorithm. Then impute $r_{iz}$ using the kNNI algorithm and the k-nearest neighbors $d_k$. Based on the imputed value $r'_{iz}$ and the actual value AV, calculate the $RMSE_k$. This has been iteratively performed $N_p - 1$ times, so $N_p - 1$ estimations and RMSE values are generated.

Step 11—Sort ascendingly the obtained $N_p - 1$ RMSE values and find minimum RMSE value.

Step 12—Determines the k value that produces the minimal RMSE value.

Step 13—Find the k-nearest neighbors $d_k$ of $R_i$ within P using kNN algorithm employing the best k value found in step 12 and create a subset S made up with $d_k$ and $R_i$.

Step 14—Feed the subset S to Iterative Imputer. Iterative Imputer imputes the missing values in $R_i$ and returns the imputed record $R'_i$.

Step 15—Remove incomplete record $R_i$ from D.

Step 16—Add imputed record $R'_i$ into D.

Step 17—Remove incomplete record $R_i$ from $D_{mis}$.

Step 18—Return iteratively to step 2 as long as $D_{mis} \neq \varnothing$ for imputing the rest of the incomplete records in $D_{mis}$.

This paper then proposes another enhanced hybrid missing data imputation method, called FCKI, that is an extension of KI. It integrates fuzzy c-means, k-nearest neighbors, and iterative imputation algorithms for imputing the missing data for a dataset. This technique focuses on improving time efficiency for the proposed missing data imputation algorithm as well as missing data imputation accuracy. The general scheme for FCKI is illustrated in Fig. 2. FCKI performs fuzzy c-means clustering for the dataset to divide records of the dataset into c fuzzy clusters where the records in the same cluster are more similar to each other. Then, it imputes each cluster separately using KI algorithm through two phases. In the first phase, the best set of nearest neighbors for each missing record is discovered based on records similarity by the kNN algorithm. Phase two focuses on exploiting the iterative imputation method to impute the missing values of the incomplete records by using the global correlation structure among the selected records. Therefore, the technique applies two levels of similarity.

For the first level of similarity, fuzzy c-means clustering is selected. The clustering approaches are often unsupervised strategies that can be utilized to break down data into sub-groups or clusters using the similarities through records (*Pinzon-Morales et al., 2011*). They are divided into two categories, named hard (crisp) clustering and fuzzy (soft)

---

**Algorithm 4  KI.**

**Input:** A dataset $D_{N*M}$ with missing values

**Output:** A dataset $D'$ with all missing values imputed

**Begin**

  $D_{mis} \leftarrow$ Find a set of records in D which contain missing values

  **while** $D_{mis} \neq \varnothing$ **do**

    $R_i \leftarrow$ Select an incomplete record from $D_{mis}$

    $A_{mis} = \{\}$

    **for** $j = 1, \ldots, M$ **do**

      **if** $r_{ij} = $ NaN **then**

        $A_{mis} = A_{mis}$ U $A_j$ /* Find a set of attributes which contain missing values in $R_i$ */

      **end**

    **end**

    $P \leftarrow$ Find a set of records in D which haven't missing values in corresponding attributes $A_{mis}$

    $P = P$ U $R_i$ /* Add the incomplete record $R_i$ into the pool P */

    $Temp = P$ /* Copy the pool P into Temp */

    $z \leftarrow$ Find a random index of the available value in $R_i$  /* Find a random index z of an attribute for which the value of $r_{iz}$ is available */

    $AV = r_{iz}$ /* Preserve the value $r_{iz} \in R_i$ into a variable of actual value (AV) */

    $r_{iz} = $ NaN /* Create missing value into $r_{iz}$ within Temp*/

    **for** $k = 2, \ldots, N_p - 1$ **do**

      $d_k \leftarrow$ FindKNNRecords($R_i$, Temp, k); /* Find the k-nearest neighbors of $R_i$ within Temp by using kNN algorithm */

      $r_{iz}' \leftarrow$ kNNI($R_i$, $d_k$, k) /* Impute $r_{iz}$ using kNNI algorithm and the k-nearest neighbors $d_k$ */

      $RMSE_k \leftarrow$ CalculateRMSE(AV, $r_{iz}'$); /* Compute RMSE value between the existing value AV and the imputed value $r_{iz}'$ */

    **end**

    $k \leftarrow$ argmin$\{RMSE_k\}$ /* Find the minimum RMSE and return corresponding k */

    $d_k \leftarrow$ FindKNNRecords($R_i$, P, k); /* Find the k-nearest neighbors of $R_i$ within P by using kNN algorithm */

    $S = d_k$ U $R_i$  /* Create a subset made up with $d_k$ and $R_i$ */

    $R_i' \leftarrow$ IterativeImputer(S) /* Impute the missing values in $R_i$ using Iterative Imputation and return $R_i'$ */

    $D = D \setminus R_i$  and $D_{mis} = D_{mis} \setminus R_i$  /* Remove incomplete record $R_i$ from D and $D_{mis}$ */

    $D = D$ U $R_i'$ /* Add imputed record $R_i'$ into D */

  **End**

  $D' = D$

  Return complete dataset $D'$

**End**

---

clustering. In the case of crisp clustering, like the k-means algorithm, a record $R_i$ assigned to one and only one cluster to which $R_i$ is the utmost similar (*Rahman & Islam, 2016*). In the case of fuzzy clustering techniques, records on the frontiers between multiple clusters are not compelled to assign to one of the clusters completely. The records can

**Table 1 The definitions of notations used in Algorithm 4.**

| Notation | Definition |
|----------|-----------|
| D | Incomplete dataset |
| D′ | Imputed dataset of D |
| N | Total number of records in D |
| M | Total number of attributes in D |
| $D_{mis}$ | Incomplete subset of D |
| $R_i$ | Incomplete target record from $D_{mis}$ |
| $R_i′$ | Imputed record of $R_i$ |
| $A_{mis}$ | Set of attributes that contain missing values in $R_i$ |
| $A_j$ | Incomplete attribute within $R_i$ |
| P | Pool that contains a set of records in D which haven't missing values in corresponding attributes $A_{mis}$ |
| Temp | Copy of the pool P |
| $r_{iz}′$ | Imputed value of $r_{iz}$ |
| $N_p$ | Number of records in P |
| $d_k$ | k nearest neighbors of $R_i$ |
| S | Subset made up of $d_k$ and $R_i$ |

belong to multiple clusters at the same time (*Sefidian & Daneshpour, 2019*). Each record has a membership degree between 0 and 1, showing its partial membership.

The main k-means is used for imputing the missing data, but fuzzy clustering has many advantages because it is more realistic contrariwise hard clustering in many situations. when records are not well-separated, as is the case for missing data problems, it provides a better description tool (*Sefidian & Daneshpour, 2019*). Besides, if the initial points are not chosen properly, the original k-means algorithm may be stuck in the local minimum state. Continuous membership values in the fuzzy clustering, on the other hand, provide the resulting algorithms minimal sensitivity to achieve stuck at a local minimum (*Li et al., 2004*; *García, Luengo & Herrera, 2015*).

The most popular soft clustering technique is the fuzzy c-Means (FCM) algorithm (*Bezdek, Ehrlich & Full, 1984*; *Sefidian & Daneshpour, 2019*). The algorithm of the FCM partitions a set of input data $\{R_1, R_2, \ldots, R_n\}$ into c fuzzy clusters $\{C_1, C_2, \ldots, C_c\}$ by minimizing the following objective function, which is based on distance:

$$\sum_{i=1}^{n} \sum_{k=1}^{c} (\delta_{ik})^{m'} \left\| R_i - c_k \right\|^2 \tag{1}$$

$R_i = [R^1, R^2, \ldots, R^m]^T$ represents an input record, and $R^j$ refers to the value of $j^{th}$ attribute for $R_i$. $c_k$ denotes the $k^{th}$ cluster prototype (centroid). $m' \in (1, \infty)$ is a fuzzification parameter that specifies how much the clusters can overlap. $\|.\|$ denotes the euclidean norm which is used to measure the similarity of data record $R_i$ to the center vector $c_k$. $\delta_{ik}$ is the likelihood value that expresses the degree to which $R_i$ belongs to the $k^{th}$ cluster ($C_k$), $\forall i$, $k: \delta_{ik} \in [0, 1]$. The increased value of $\delta_{ik}$ expresses the increased association between $R_i$ and $C_k$. The total association of $R_i$ with c clusters is equal to 1.

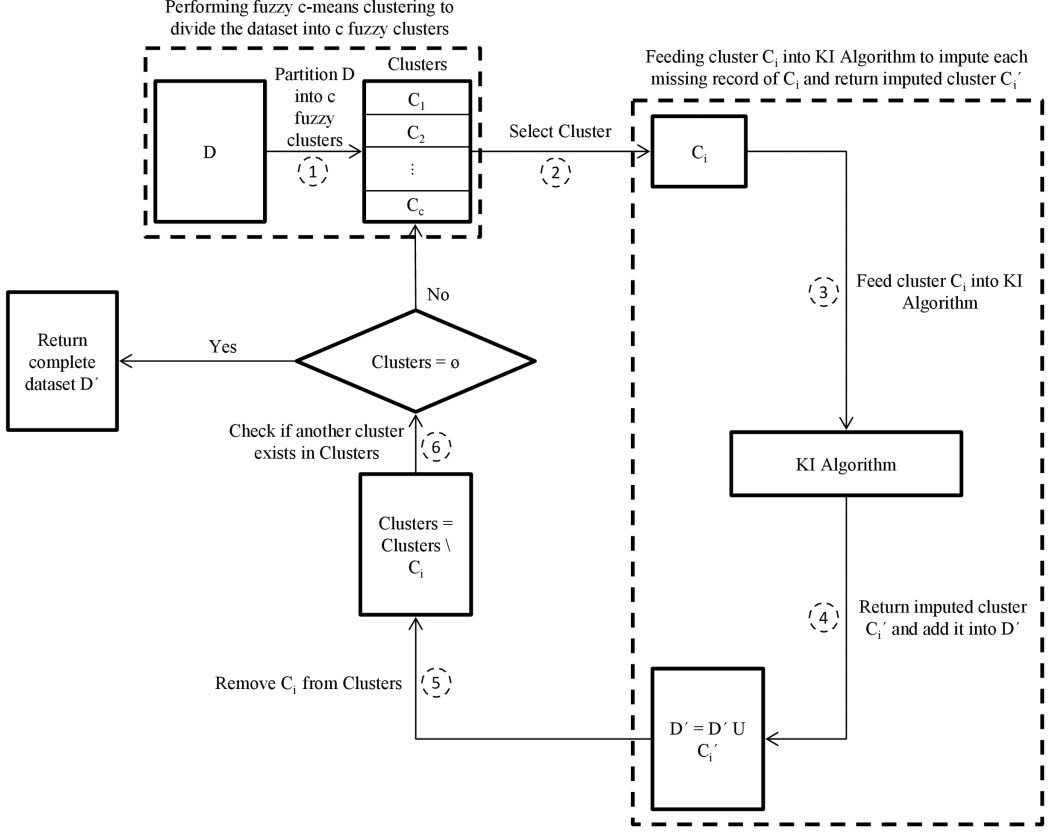

**Figure 2  The proposed FCKI scheme.** D is the incomplete dataset, D′ is the imputed dataset of D, c is the optimal number of clusters in dataset D, Clusters is the fuzzy clusters that partitioned using fuzzy c-means algorithm, $C_i$ is the target cluster from Clusters, $C_i′$ is the imputed cluster of $C_i$.

Based on fuzzy set theory, incomplete records may belong to multiple clusters at the same time. A record $R_i$ with a missing value is considered to have a membership degree (fuzzy association) with each cluster. The cluster with which the record $R_i$ has a higher membership degree has a greater effect on the imputation than the cluster with the lower membership degree. So when a missing record $R_i$ belongs to two or more clusters, FCKI get imputed record $R_i′$ only from the cluster which has highest membership degree.

In cluster analysis, the elbow approach is a heuristic utilized in deciding the number of clusters in a dataset. The approach comprises plotting the demonstrated difference as a function of the number of clusters and picking the curve elbow as the number of clusters to be used (*Kodinariya & Makwana, 2013*).

For the second level of similarity, kNN is selected. It finds the best k records that are the utmost similar to the missing record by using the Euclidean distance measure. FCKI, similar to KI finds the best k value for the kNN automatically.

The proposed imputation method has the advantage of tackling missing values based on the similarity of the set of records instead of the whole dataset. Iterative imputation is applied to discover the global correlation among the selected records that are similar to each other to impute missing values. It does not have many iterations to impute missing

---

**Algorithm 5  FCKI.**

**Input:** A dataset $D_{N*M}$ with missing values

**Output:** A dataset D′ with all missing values imputed

**Begin**

  c = Elbow(D) /* Determine the optimal number of clusters in dataset D using elbow method */

  Clusters = FCM(D, c) /* Partition dataset D into c fuzzy clusters {$C_1$, $C_2$,…, $C_c$} using fuzzy c-means algorithm */

  **foreach** Cluster $C_i$ ∈ Clusters **do**

    $C_i'$ = KI($C_i$) /* Feed cluster $C_i$ into KI algorithm to impute each missing record of $C_i$ and return imputed cluster $C_i'$*/

    D′ = D′ U $C_i'$  /* Add imputed cluster $C_i'$ into D′ */

  **End**

  Return complete dataset D′

**End**

---

**Table 2 The definitions of notations used in Algorithm 5.**

| Notation | Definition |
|---|---|
| D | Incomplete dataset |
| D′ | Imputed dataset of D |
| N | Total number of records in D |
| M | Total number of attributes in D |
| c | Optimal number of clusters in dataset D |
| Clusters | Fuzzy clusters that partitioned using fuzzy c-means algorithm |
| $C_i$ | Target cluster from Clusters |
| $C_i'$ | Imputed cluster of $C_i$ |

values of the dataset, which is a significant factor for improving time efficiency. It focuses on both the records' similarity and the correlations through the features. Therefore, it can enhance imputation effectiveness and efficiency.

The pseudo-code of FCKI is presented in Algorithm 5. The definitions of notations used in Algorithm 5 are presented in Table 2. The main steps of the FCKI algorithm are explained in the following:

Step 1—Determine the optimal number of clusters c in dataset D using the elbow method then partition dataset D into c fuzzy clusters {$C_1$, $C_2$,…, $C_c$} using the fuzzy c-means algorithm.

Step 2—Select a cluster $C_i$ from Clusters.

Step 3—Feed cluster $C_i$ into KI algorithm to impute each missing record of $C_i$ and return imputed cluster $C_i'$.

Step 4—Add imputed cluster $C_i'$ into D′.

Step 5—Remove cluster $C_i$ from Clusters.

Step 6—Return iteratively to step 2 as long as Clusters ≠ ⊘ for imputing the rest of the incomplete clusters.

**Table 3 The datasets used in these experiments.**

| Datasets | Number of records | Number of numerical features | Dataset size (KB) |
|---|---|---|---|
| Zoo | 101 | 17 | 3.49 |
| Iris | 150 | 4 | 2.36 |
| Sonar | 208 | 60 | 83.8 |
| Glass | 214 | 9 | 9.80 |
| Ecoli | 336 | 7 | 11.2 |
| Leaf | 340 | 16 | 40.4 |
| Ionosphere | 351 | 34 | 73.3 |
| Movement libras | 360 | 90 | 250 |
| QSAR fish toxicity | 908 | 7 | 30.3 |
| Yeast | 1,484 | 8 | 52.8 |
| Abalone | 4,177 | 8 | 183 |
| Anuran Calls (MFCCs) | 7,195 | 22 | 1,780 |
| Letter | 20,000 | 16 | 676 |
| Sensorless Drive Diagnosis | 58,509 | 48 | 23,800 |
| Pseudo Periodic Synthetic | 100,000 | 10 | 11,500 |

# RESULTS AND DISCUSSION

## Experimental setup

The experiments are performed on machine 1, which is configured with 2 × 4 core Intel i7-7500U processor and 8 GB RAM. The source codes used in this work for all the various imputation methods were implemented in Python 3.8.0 with the help of some Scikit-Learn packages (*Pedregosa et al., 2011*).

## Dataset description

The proposed missing data imputation methods, KI, and FCKI, are evaluated on fifteen datasets, which are shown in Table 3. These datasets are found in UCI Machine Learning Repository. These datasets are commonly used in related work. These datasets are selected according to three factors. The first one is that these datasets were used in many previous and related works, therefore can be used for the comparison. The second factor is that these datasets have no missing values, so we can generate incomplete datasets. This is critical for assessing accuracy. Third factor is that these datasets are different in volume and the number of instances and attributes. The evaluation depends on comparing KI and FCKI with 10 different methods of missing data imputation. These ten methods are mean imputation (*Ravi & Krishna, 2014*), kNNI (*Batista & Monard, 2003*; *Rahman & Islam, 2013b*; *Liu et al., 2015*), SoftImpute (*Mazumder, Hastie & Tibshirani, 2010*), SVDImpute (*Troyanskaya et al., 2001*), traditional iterative imputation (*Little & Rubin, 2002*; *Van Buuren & Groothuis-Oudshoorn, 2011*; *Pedregosa et al., 2011*), EMI (*Schneider, 2001*; *Junninen et al., 2004*), DMI (*Rahman & Islam, 2011*), KDMI (*Rahman & Islam, 2013b*), KEMI (*Razavi-Far et al., 2020*) and KEMI$^+$ (*Razavi-Far et al., 2020*). The selected datasets do not contain realistic missing values. Algorithms are used to generate missing values in a predefined ratio for selected datasets to simulate the various types of

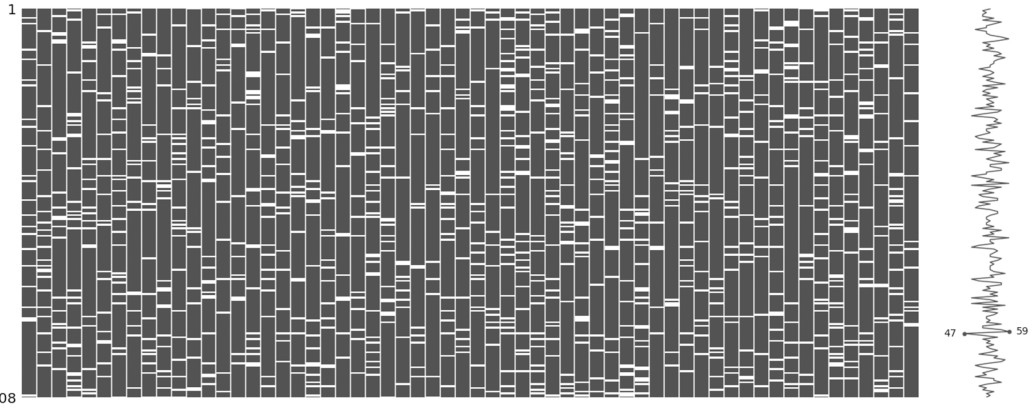

**Figure 3** **The visualization of the sonar dataset with MCAR missing data type and 10% missing ratio.**

missing data; MCAR, MAR, MNAR were provided in "Proposed Missing Data Imputation Methods". These algorithms randomly generate missing data for achieving results as close as possible to reality. The dataset positions to be modified are selected randomly based on the specified criterion for each missing data type. The missing values are generated for each dataset with different missing ratios ranging from 1% to 20% of total attribute values for each type of missing data; MCAR, MAR, MNAR. These lead to the twelve different collections generated missing data. These collections are derived from multiplying four missing ratios and three missing types over each dataset. The evaluation of the proposed methods is applied over these generated 180 different datasets, derived from multiplying fifteen datasets and twelve missing collections. The evaluation depends on using three different imputation error measures, which are the root mean square error (RMSE), the normalized root mean square error (NRMSE), and the mean absolute error (MAE).

The incomplete sonar dataset's visualization with MCAR missing data type and 10% missing ratio is illustrated in Fig. 3. This visualization can assist in showing missing and non-missing values in the dataset. The non-missing values are represented by black cells, and the missing values are represented by white cells. The graph shown in Fig. 3 was generated using a python library called missingno (*Bilogur, 2018*). This library provides the ability to know how missing values are distributed via informative visualizations. The number of missing values in each feature for an incomplete sonar dataset with MCAR missing data type and 10% missing ratio is illustrated in Fig. 4. The missing data is MCAR when missing in the dataset occurs entirely at random, and there is no specific pattern can be determined as shown in Fig. 3. The incomplete sonar dataset's visualization through MAR missing data type and 10% missing ratio is illustrated in Fig. 5. The number of missing values in each feature for an incomplete sonar dataset with MAR missing data type and 10% missing ratio is illustrated in Fig. 6. The missing data is MAR when a specific pattern can be determined. The probability that the value of a particular variable is missed or not for any observation is depending on the values of other variables,
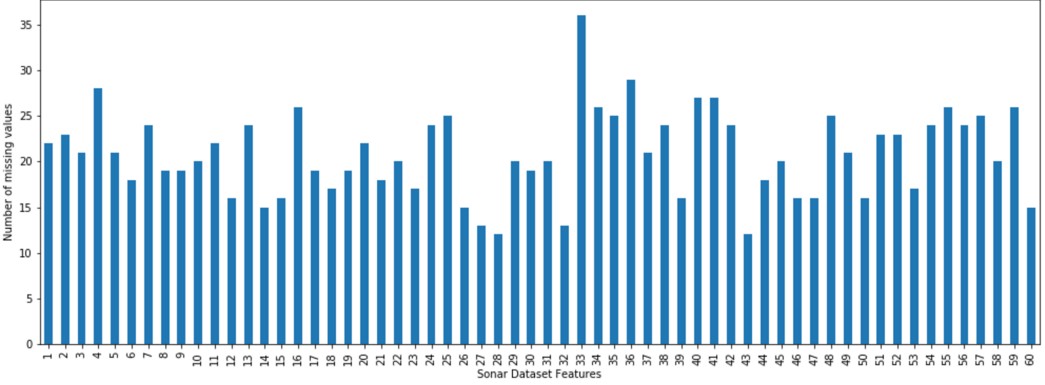

**Figure 4** The number of missing values in each feature for an incomplete sonar dataset with MCAR missing data type and 10% missing ratio.

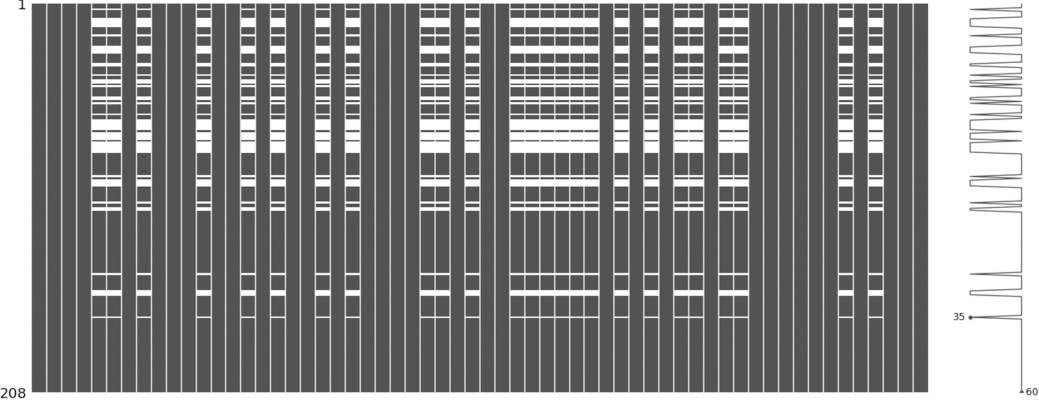

**Figure 5** The visualization of sonar dataset with MAR missing data type and 10% missing ratio.

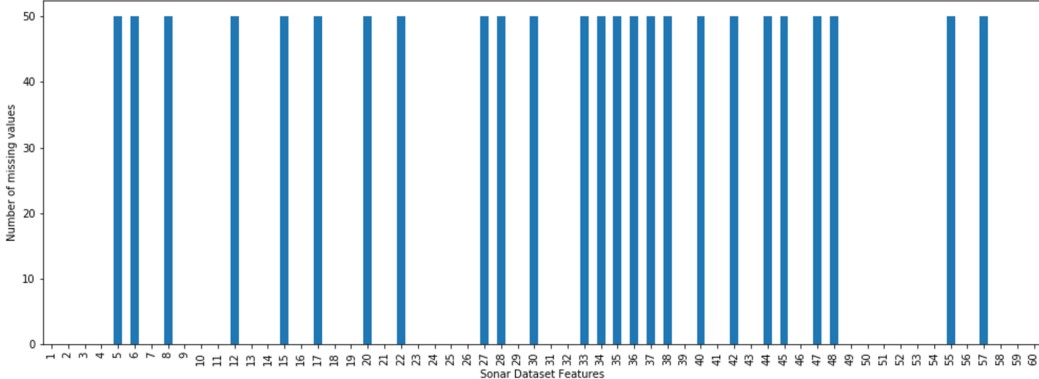

**Figure 6** The number of missing values in each feature for an incomplete sonar dataset with MAR missing data type and 10% missing ratio.

so a common factor can be found in all observations that have missing values. The representation of the incomplete sonar dataset with MNAR missing data type and 10% missing ratio is illustrated in Fig. 7. The number of missing values in each feature for

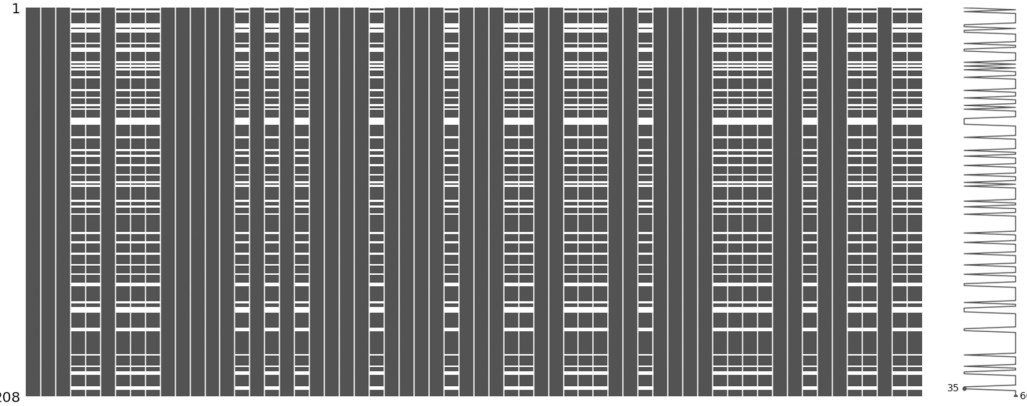

**Figure 7  The visualization of sonar dataset with MNAR missing data type and 10% missing ratio.**

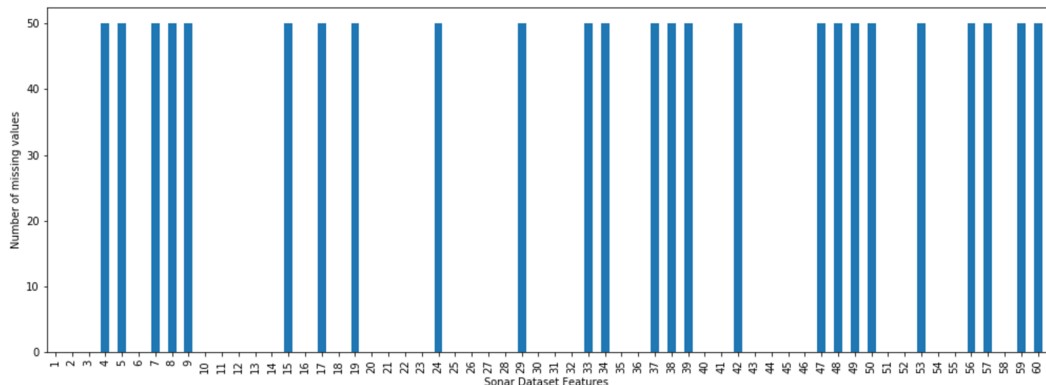

**Figure 8  The number of missing values in each feature for an incomplete sonar dataset with MNAR missing data type and 10% missing ratio.**           

an incomplete sonar dataset with MNAR missing data type and 10% missing ratio is shown in Fig. 8. The missing data is MNAR when the data that cause others to be missing are unobserved.

## Evaluation criteria

The methods which are chosen as competitors are Mean Imputation, kNNI, SoftImpute, SVDimpute, Iterative Imputation, EMI, DMI, KDMI, KEMI and KEMI[+]. These methods are used because the literature review mentioned that there is a need to be applied for many applications with different sizes of datasets and they have been shown to be effective. Some recent research relied on these methods for comparison. Moreover, most of these methods rely on data similarity, which is the basis of the proposed methods. These methods gradually vary from simple to advanced. These methods are compared to the proposed missing data imputation methods, KI, and FCKI in terms of three imputation performance measures; the root mean square error (RMSE), the normalized root mean

**Table 4 The average RMSE values over twelve experiments were obtained for each dataset.**

| Datasets | Mean | kNNI | SoftImpute | SVDimpute | Iterative Imputation | EMI | DMI | KDMI | KEMI | KEMI+ | KI | FCKI |
|---|---|---|---|---|---|---|---|---|---|---|---|---|
| Zoo | 0.4457 | 0.2483 | 0.2310 | 0.2575 | 0.1837 | 0.2219 | 0.1935 | 0.1766 | 0.1077 | 0.0965 | **0.0757** | **0.0632** |
| Iris | 0.5946 | 0.2878 | 0.3338 | 0.4114 | 0.1866 | 0.3818 | 0.3333 | 0.3044 | 0.1525 | 0.1369 | **0.0903** | **0.0745** |
| Sonar | 0.0916 | 0.0571 | 0.0289 | 0.0267 | 0.0289 | 0.0471 | 0.0410 | 0.0375 | 0.0302 | 0.0271 | **0.0177** | **0.0194** |
| Glass | 0.4818 | 0.3482 | 0.3694 | 0.2660 | 0.2314 | 0.2906 | 0.2173 | 0.1982 | 0.1466 | 0.1314 | **0.1076** | **0.1120** |
| Ecoli | 0.0797 | 0.0694 | 0.0550 | 0.0449 | 0.0455 | 0.0419 | 0.0363 | 0.0332 | 0.0277 | 0.0248 | **0.0208** | **0.0223** |
| Leaf | 0.2031 | 0.1717 | 0.1625 | 0.1660 | 0.1544 | 0.1775 | 0.1509 | 0.1358 | 0.1181 | 0.1074 | **0.0642** | **0.0546** |
| Ionosphere | 0.3264 | 0.2392 | 0.1464 | 0.1558 | 0.1937 | 0.1613 | 0.1404 | 0.1282 | 0.0984 | 0.0884 | **0.0701** | **0.0683** |
| Movement libras | 0.1012 | 0.0444 | 0.0244 | 0.0624 | 0.0479 | 0.0520 | 0.0452 | 0.0413 | 0.0289 | 0.0259 | **0.0131** | **0.0123** |
| QSAR fish toxicity | 0.4263 | 0.3573 | 0.3091 | 0.3141 | 0.2530 | 0.2923 | 0.2244 | 0.2159 | 0.2003 | 0.1926 | **0.1354** | **0.1258** |
| Yeast | 0.0655 | 0.0580 | 0.0392 | 0.0308 | 0.0353 | 0.0288 | 0.0250 | 0.0228 | 0.0225 | 0.0201 | **0.0137** | **0.0137** |
| Abalone | 0.4009 | 0.2534 | 0.2370 | 0.2751 | 0.1677 | 0.2017 | 0.1756 | 0.1606 | 0.0882 | 0.0791 | **0.0353** | **0.0329** |
| Anuran Calls (MFCCs) | 0.0868 | 0.0373 | 0.0382 | 0.0359 | 0.0293 | 0.0437 | 0.0380 | 0.0347 | 0.0252 | 0.0226 | **0.0151** | **0.0133** |
| Letter | 0.6320 | 0.5476 | 0.4684 | 0.5444 | 0.3794 | 0.6101 | 0.5328 | 0.4864 | 0.3546 | 0.3180 | **0.1606** | **0.1416** |
| Sensorless Drive Diagnosis | 0.1019 | 0.0574 | 0.0420 | 0.1366 | 0.0643 | 0.0885 | 0.0524 | 0.0478 | 0.0346 | 0.0310 | **0.0208** | **0.0183** |
| Pseudo Periodic Synthetic | 0.0597 | 0.1327 | 0.0534 | 0.0348 | 0.0305 | 0.0302 | 0.0263 | 0.0240 | 0.0175 | 0.0157 | **0.0105** | **0.0092** |
| **Standard Deviation** | 0.2009 | 0.1450 | 0.1420 | 0.1525 | 0.1022 | 0.1588 | 0.1370 | 0.1255 | 0.0887 | 0.0807 | **0.0472** | **0.0430** |

Note:
KI and FCKI results are shown in bold.

square error (NRMSE), and the mean absolute error (MAE) which are given by the following equations:

$$\text{RMSE} = \sqrt{\frac{1}{N}\sum_{i=1}^{N}(P_i - O_i)^2} \tag{2}$$

$$\text{NRMSE} = \frac{\sqrt{\frac{1}{N}\sum_{i=1}^{N}(P_i - O_i)^2}}{O_{max} - O_{min}} \tag{3}$$

$$\text{MAE} = \frac{1}{N}\sum_{i=1}^{N}|P_i - O_i| \tag{4}$$

P and O are the predicted value and the observed value, respectively. N is the total number of records. $O_{max}$ and $O_{min}$ are the maximum observed value and the minimum observed value, respectively.

## Results

The average values of RMSE obtained over twelve experiments for each dataset utilizing each imputation method are shown in Table 4. The best imputation method is the one with the lowest RMSE value among all of these imputation methods. The results show that FCKI followed by KI performs significantly better than mean, kNNI, SoftImpute, SVDimpute, Iterative Imputation, EMI, DMI, KDMI, KEMI, and KEMI[+]. The reported results in Table 4 show that FCKI and KI have the lowest average RMSE among the other

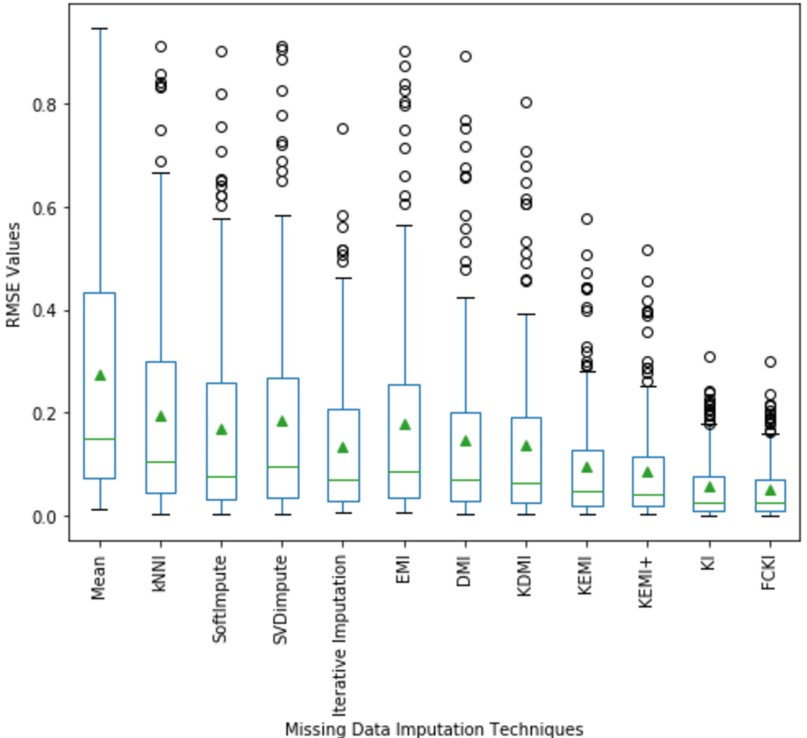

**Figure 9 Distribution of the RMSE values obtained by each method.**

imputation methods for all datasets (in Table 4, see the bold entries). The worst result was obtained by mean imputation. This is because the mean imputation entirely ignores the correlations between the features. Standard deviation of the average values of RMSE for each imputation method is presented in the last row of Table 4. In statistics, standard deviation is a measure of distribution. It is used to determine the spread and variation of a series of data values. A low standard deviation value means that the data are less spread out, while a high standard deviation value indicates that the data in a collection are distributed apart from their mean average values. The reported results of standard deviation in Table 4 show that FCKI and KI have the lowest standard deviation among the other imputation methods. This indicates that FCKI and KI methods can provide stable imputation results over different datasets. The average NRMSE and MAE values derived from twelve experiments for each dataset utilizing each imputation method are presented in Appendix A as Tables A1 and A2, respectively.

A boxplot of the imputation performance evaluation obtained over the datasets mentioned in Table 3 with various missing data types; MCAR, MAR, MNAR, and missing ratios; 1%, 5%, 10%, 20% is illustrated in Fig. 9. The box part of the boxplot is described by two lines at the 25th percentile and 75th percentile. The 25th percentile is the value at which 25% of the RMSE values are less than this value. Thus, the middle 50% of the RMSE values fall between the 25th percentile and the 75th percentile. The distance between the upper and lower lines of the box is called the interquartile range (IQR), which is a common measure of the spread of the RMSE values. A line inside the box is the

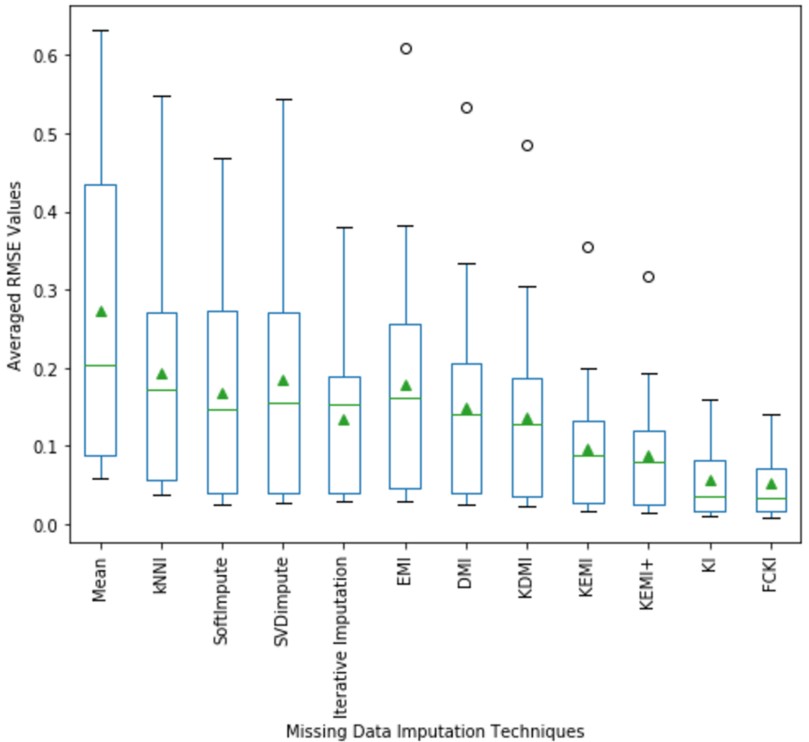

**Figure 10 The average value of RMSE values and their distribution are obtained by each method.**

median. The solid dashes represent the 1st and 99th percentiles. The circles represent outliers. The solid triangles represent the averaged RMSE value obtained by each imputation method. The 180 RMSE values obtained by each technique are in each box. Figure 9 shows that the medians of the RMSE values for FCKI and KI methods are lower than those of the other ten imputation methods, which indicates that FCKI and KI methods have less error for missing data imputation. In practice, the method of missing data imputation with a strong statistical property is that it has a lower RMSE. The distances between the 25th and 75th percentiles of RMSE values for FCKI and KI methods are the smallest and the normal ranges of RMSE values for FCKI and KI methods are also the smallest, which shows that FCKI and KI methods can provide stable imputation results over different datasets with various missing data types; MCAR, MAR, MNAR, and missing ratios; 1%, 5%, 10%, 20%. The solid triangles indicate that FCKI and KI have the lowest average overall RMSE values. The results show that FCKI and KI outperform other imputation methods, and they are the most stable techniques. Boxplots for distribution of NRMSE and MAE values obtained over the datasets mentioned in Table 3 with various missing data types; MCAR, MAR, MNAR, and missing ratios; 1%, 5%, 10%, 20% are presented in Appendix B as Figs. B1 and B2, respectively.

The average values of RMSE and its distribution for all experiments implemented on all used datasets mentioned in Table 3 are illustrated in Fig. 10. The fifteen average values of RMSE obtained by each technique are in each box. The average value of RMSE values also indicates that FCKI and KI outperform other imputation techniques.

**Table 5 The average value of RMSE values for all datasets achieved by applying each imputation method to each missing data type.**

| Datasets | Mean | kNNI | SoftImpute | SVDimpute | Iterative imputation | EMI | DMI | KDMI | KEMI | KEMI[+] | KI | FCKI |
|---|---|---|---|---|---|---|---|---|---|---|---|---|
| MCAR | 0.3204 | 0.2279 | 0.1912 | 0.2186 | 0.1565 | 0.2176 | 0.1769 | 0.1625 | 0.1153 | 0.1047 | **0.0686** | **0.0612** |
| MAR | 0.2612 | 0.1842 | 0.1519 | 0.1709 | 0.1277 | 0.1575 | 0.1358 | 0.1232 | 0.0906 | 0.0819 | **0.0527** | **0.0492** |
| MNAR | 0.2378 | 0.1699 | 0.1646 | 0.1629 | 0.1222 | 0.1587 | 0.1338 | 0.1239 | 0.0847 | 0.0770 | **0.0489** | **0.0459** |

Note:
KI and FCKI results are shown in bold.

**Table 6 The average value of RMSE values for all datasets achieved by applying each imputation method to each missing ratio.**

| Datasets | Mean | kNNI | SoftImpute | SVDimpute | Iterative Imputation | EMI | DMI | KDMI | KEMI | KEMI[+] | KI | FCKI |
|---|---|---|---|---|---|---|---|---|---|---|---|---|
| 1% | 0.1033 | 0.0655 | 0.0639 | 0.0657 | 0.0470 | 0.0643 | 0.0538 | 0.0492 | 0.0341 | 0.0310 | **0.0193** | **0.0176** |
| 5% | 0.2292 | 0.1412 | 0.1359 | 0.1549 | 0.0991 | 0.1444 | 0.1210 | 0.1109 | 0.0768 | 0.0697 | **0.0436** | **0.0398** |
| 10% | 0.3382 | 0.2317 | 0.1965 | 0.2259 | 0.1602 | 0.2121 | 0.1774 | 0.1630 | 0.1161 | 0.1050 | **0.0678** | **0.0618** |
| 20% | 0.4219 | 0.3375 | 0.2807 | 0.2902 | 0.2354 | 0.2910 | 0.2431 | 0.2228 | 0.1605 | 0.1457 | **0.0962** | **0.0892** |

Note:
KI and FCKI results are shown in bold.

The boxplots of the average value of NRMSE and MAE values and its distribution for all experiments implemented on all datasets mentioned in Table 3 are presented in Appendix C as Figs. C1 and C2, respectively.

The average value of RMSE values for all datasets achieved by applying each imputation method to each missing data type are shown in Table 5. The results show that FCKI and KI outperform other imputation methods for all missing data types (in Table 5, see the bold entries). The average value of NRMSE and MAE values for all datasets achieved by applying each imputation method to each missing data type are presented in Appendix D as Table D1 and D2, respectively.

The average value of RMSE values for all datasets achieved by applying each imputation method to each missing ratio are shown in Table 6. The results show that FCKI and KI outperform other imputation methods for all missing ratios (in Table 6, see the bold entries). The average value of NRMSE and MAE values for all datasets achieved by applying each imputation method to each missing ratio are presented in Appendix E as Table E1 and E2, respectively.

The average value of RMSE values for MCAR, MAR, and MNAR missing data types, respectively, overall datasets achieved by applying each imputation method to each missing ratio are illustrated in Figs. 11–13. The lowest averages of RMSE values are achieved by applying FCKI and KI, and thus they outperform the other methods of imputation, as shown in Figs. 11–13. The average value of NRMSE and MAE values for MCAR, MAR, and MNAR missing data types, respectively, for datasets achieved by applying each imputation method to each missing ratio, are presented in Appendix F as Figs. F1–F3 and Figs. F4–F6, respectively.

The missing data imputation methods are tested in order to compare them statistically. First, a Friedman rank test with a significance level $\alpha = 0.05$ is performed. This test is used to compare the variations between these imputation methods. It decides if one or
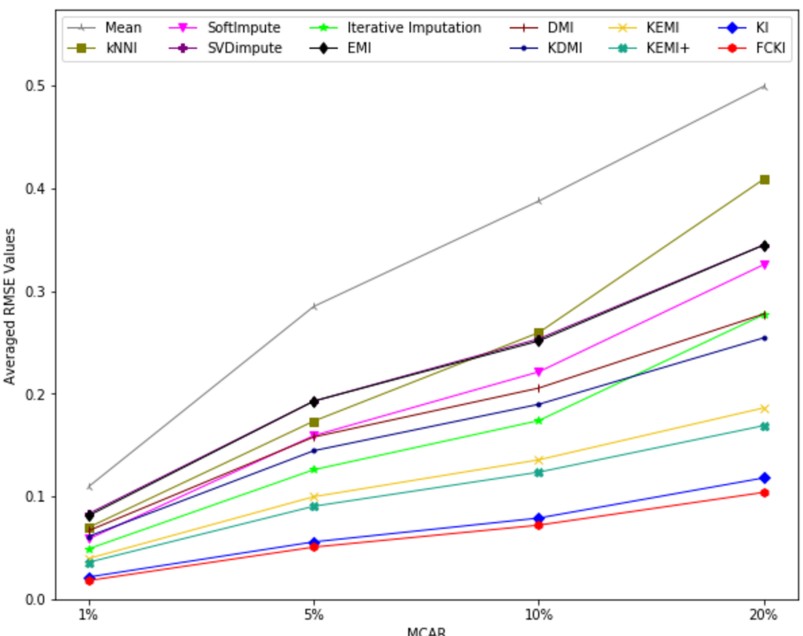

**Figure 11** The average value of RMSE values for MCAR missing data type overall datasets achieved by applying each imputation method to each missing ratio.

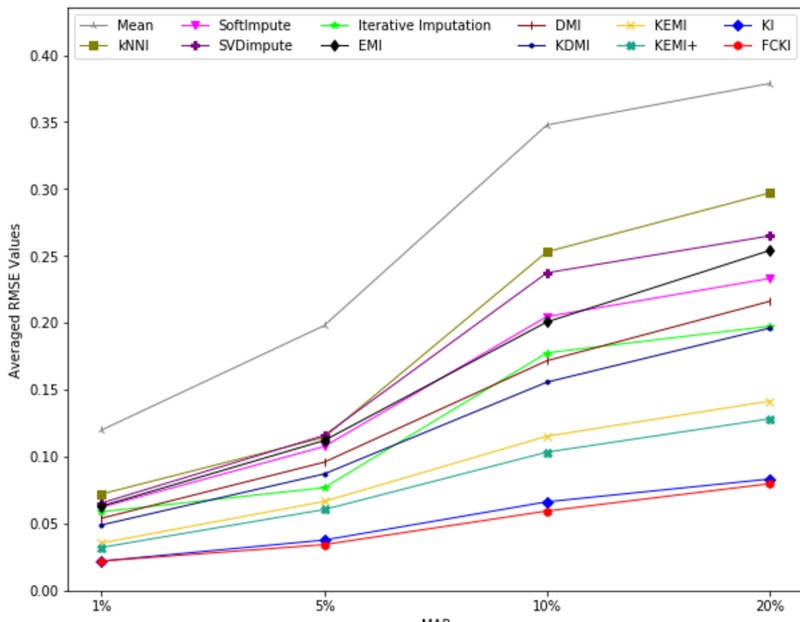

**Figure 12** The average value of RMSE values for MAR missing data type for datasets achieved by applying each imputation method to each missing ratio.

more imputation methods have significantly different performance. The test statistic is 141.420 and the corresponding p-value is 0 for the obtained RMSE. Since the p-value is less than the significance level of 0.05, the null hypothesis is rejected, and it is concluded

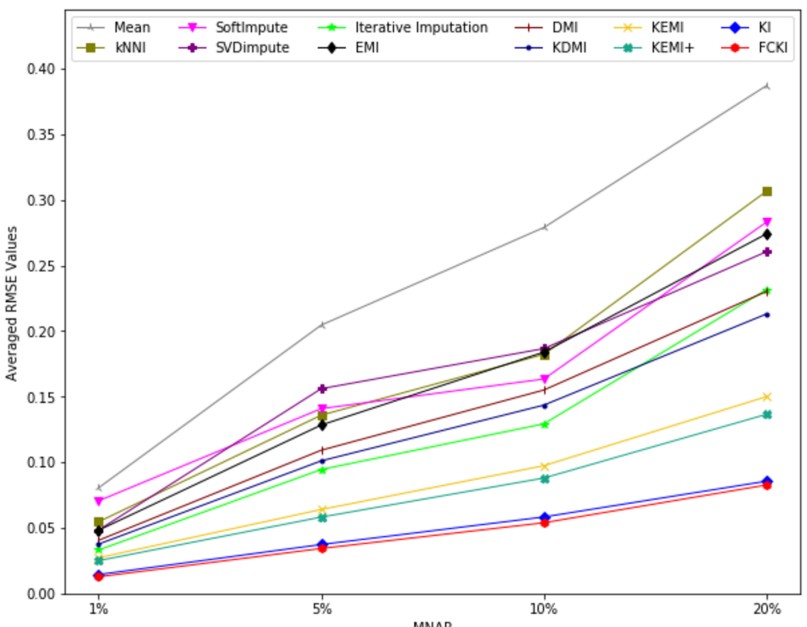

**Figure 13 The average value of RMSE values for MNAR missing data type for datasets achieved by applying each imputation method to each missing ratio.**

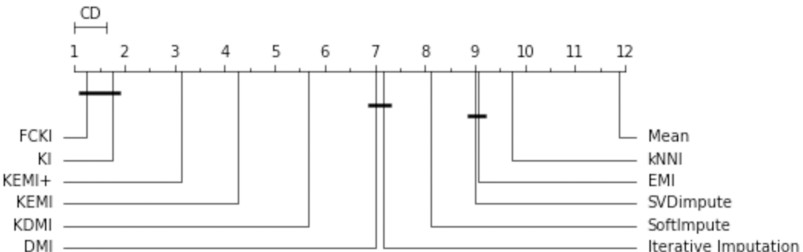

**Figure 14 The critical difference for the Nemenyi test.**

that at least one of these strategies has a different effect. As a result, a post-hoc test can be used to further assess the variations statistically. This is used to compare all the algorithms in a pairwise manner, and it is based on the absolute difference between the imputation methods' average rankings. For a significance level α = 0.05, the critical difference (CD) is 0.64. The null hypothesis that two approaches have the same performance is rejected if the difference between their average rankings is greater than CD. A diagram of the critical difference for the post-hoc Nemenyi test (*Gardner & Brooks, 2017*) is shown in Fig. 14. It compares all imputation methods with each other in terms of RMSE. It illustrates the average rank of each imputation method, wherever the first rank represents the method with the lowest RMSE. The methods are connected by thick lines if they are not significantly different. This figure shows that KEMI[+] is around the third rank, KEMI is around the fourth rank. They are followed by KDMI, DMI, Iterative Imputation, SoftImpute, SVDimpute, EMI, kNNI, and Mean. It shows that mean is the

**Table 7  The average execution times (in seconds) over 12 experiments achieved by using each dataset of the 15 datasets for advanced methods.**

| Datasets | DMI | KDMI | KEMI | KEMI$^+$ | KI | FCKI |
|---|---|---|---|---|---|---|
| Zoo | 6.9302 | 15.4543 | 7.4657 | 20.4393 | 10.3779 | **4.3305** |
| Iris | 5.9100 | 13.1793 | 11.9682 | 30.2816 | 13.7762 | 3.8477 |
| Sonar | 83.5382 | 186.2902 | 65.1894 | 193.8505 | 169.6246 | 53.4663 |
| Glass | 34.8523 | 77.7207 | 52.9551 | 145.4900 | 70.0241 | 23.1727 |
| Ecoli | 90.0214 | 200.7477 | 125.9442 | 280.3394 | 153.6133 | 59.6401 |
| Leaf | 46.0662 | 102.7276 | 139.3200 | 348.8740 | 210.7502 | 30.1334 |
| Ionosphere | 160.7483 | 358.4687 | 237.8846 | 583.1687 | 291.7119 | 103.0051 |
| Movement libras | 510.6320 | 1138.7093 | 384.2715 | 1,174.0173 | 861.1263 | 320.5707 |
| QSAR fish toxicity | 279.0096 | 622.1914 | 2,809.2265 | 4,340.2099 | 2,645.1605 | 194.9056 |
| Yeast | 1,817.4238 | 4,052.8550 | 8,701.8106 | 17,678.8337 | 9,117.6113 | 1,196.9684 |
| Abalone | 5,077.1513 | 11,322.0474 | 24,401.2329 | 29,223.1462 | 25,279.0786 | 3,352.4283 |
| Anuran Calls (MFCCs) | 8,779.8995 | 19,579.1758 | 41,981.3752 | 50,185.9836 | 43,475.5194 | 5,786.6943 |
| Letter | 24,335.2487 | 54,267.6046 | 116,691.7426 | 139,713.4244 | 120,909.3370 | 16,048.3162 |
| Sensorless Drive Diagnosis | 71,623.0829 | 159,719.4750 | 342,999.0374 | 411,265.5116 | 355,690.0018 | 47,175.6676 |
| Pseudo Periodic Synthetic | 122,172.0769 | 272,443.7315 | 585,583.7128 | 702,675.4554 | 607,250.3102 | **80,491.5812** |

**Note:**
The minimum and the maximum average execution times among the datasets for the FCKI algorithm are shown in bold.

least accurate method. This figure shows that FCKI and KI outperform significantly other methods.

The evolution of the runtimes, which is studied concerning the dataset's volume, is considered another matter of concern. The experiments investigate the scalability of the missing data imputation methods by assessing runtimes for all datasets with different missing ratios ranging from 1% to 20% of total attribute values for each type of missing data; MCAR, MAR, MNAR. The type of time that is measured for these experiments is wall-clock time. The wall-clock time measures the total time to execute a program on a computer. This time is measured using a python module named time. The paper only shows the runtimes for advanced techniques; DMI, KDMI, KEMI, KEMI$^+$, KI, FCKI. They take more time than other traditional techniques to pay the price of a significantly better quality of imputation. The average execution times (in seconds) achieved over twelve experiments for each dataset utilizing each advanced imputation method are shown in Table 7. The average execution times of FCKI which are presented in Table 7 include the clustering process but do not include finding the best c value. The results show that FCKI performs significantly better than DMI, KDMI, KEMI, KEMI$^+$ and KI. Table 7 Indicates that FCKI has the lowest average execution time among the other advanced imputation techniques for all datasets. The bold entries in Table 7 show the minimum and the maximum average execution times among the datasets for FCKI algorithm. The minimum average execution time is obtained by Zoo dataset. This time is very short because the dataset has only 101 records and 17 features. The maximum average execution time is obtained by Pseudo Periodic Synthetic dataset. This time is large because the dataset has 100,000 records and ten features. However, FCKI takes significantly less

**Table 8 The Average execution times (in seconds) overall datasets achieved by applying each advanced imputation method to each missing type.**

| Datasets | DMI | KDMI | KEMI | KEMI$^+$ | KI | FCKI |
|---|---|---|---|---|---|---|
| MCAR | 25,446.1030 | 56,744.8098 | 103,294.8530 | 124,868.2018 | 10,7286.7907 | **16,553.4577** |
| MAR | 10,634.5571 | 23,715.0623 | 65,024.9285 | 78,416.2668 | 67,353.9784 | **7,084.7802** |
| MNAR | 10,923.8582 | 24,360.2037 | 56,518.8458 | 68,287.3365 | 58,588.8357 | **7,330.7078** |

Note:
Results for FCKI are in bold.

**Table 9 The Average execution times (in seconds) overall datasets achieved by applying each advanced imputation method to each missing ratio.**

| Datasets | DMI | KDMI | KEMI | KEMI$^+$ | KI | FCKI |
|---|---|---|---|---|---|---|
| 1% | 4,250.1756 | 9,477.8915 | 23,609.3678 | 28,534.4288 | 24,501.2017 | **1,847.9024** |
| 5% | 15,623.3074 | 34,839.9754 | 63,508.3529 | 76,767.5670 | 65,889.8793 | **8,222.7933** |
| 10% | 21,334.9199 | 47,576.8715 | 94,694.0024 | 114,281.9859 | 98,208.6669 | **13,334.3250** |
| 20% | 21,464.2882 | 47,865.3626 | 117,973.1133 | 142,511.7585 | 122,373.0583 | **17,886.9068** |

Note:
Results for FCKI are in bold.

average execution time than other advanced imputation techniques. The Average execution times (in seconds) overall datasets achieved by applying each advanced imputation method to each missing data type are shown in Table 8. The results indicate that FCKI outperforms the other advanced imputation methods for all missing data types. The average execution times (in seconds) overall datasets achieved by applying each advanced imputation method to each missing ratio are shown in Table 9. The results indicate that FCKI outperforms the other advanced imputation methods for all missing ratios.

## Discussion

In this paper, two missing data imputation methods are proposed, named KI and FCKI, that aim to learn both similarities between the records and correlation among the features within the dataset. Improving the similarity among records can result in improving the imputation performance. This can further be improved considering the correlation among the features. KI applies only one level of similarity using the kNN algorithm. To improve the similarity, KI estimates a suitable k value automatically for the kNN by creating a random missing value $r_{iz}$ for the missing record $R_i$ that includes an actual missing value $r_{ij}$. For each possible k value, kNN finds the k-most similar records to $R_i$. The algorithm then imputes the missing value $r_{iz}$ through k-nearest neighbors using kNNI. The algorithm then calculates the root mean squared error (RMSE) using the imputed value and the actual value of $r_{iz}$. The best value of k is extracted from the set of k-nearest neighbors that deliver the minimal value of RMSE. The algorithm then Finds the k-nearest neighbors of missing record $R_i$ using kNN algorithm employing the best k value found in the previous step. The iterative imputation method is then used to impute the missing values of the missing record $R_i$ by using the global correlation structure among the selected records. FCKI differs from KI, in that FCKI applies two levels of similarity to achieve a higher

imputation accuracy before imputing the missing values through the iterative imputation. For the first level of similarity, fuzzy c-means clustering is selected. The similarities of all records belonging to the cluster are higher than the similarities of all the whole dataset records. Fuzzy c-means algorithm is selected because the records can belong to multiple clusters at the same time. This can lead to further improvement for similarity. For the second level of similarity, kNN is selected. It finds the best k records that are the utmost similar to the missing record by using the Euclidean distance measure. FCKI, similar to KI finds the best k value for the kNN automatically.

The results show that FCKI followed by KI performs significantly better than mean, kNNI, SoftImpute, SVDimpute, Iterative Imputation, EMI, DMI, KDMI, KEMI, and KEMI[+]. The reported results in Table 4 show that FCKI and KI have the lowest averaged RMSE among the other imputation methods for all the datasets mentioned in Table 3. The average RMSE values in KI and FCKI are better than KEMI[+,] which is the best competitor by 35.42% and 40.69%, respectively. In the case of MCAR missing data type, the proposed imputation techniques, KI and FCKI, can impute missing values in a dataset that does not contain any complete record (see Fig. 3). They only obtain records that have missing values then impute each incomplete record using the set of records that haven't missed values in corresponding attributes containing missing values in the required incomplete record. This also increases the number of records that can be similar to the required incomplete record that will be used for imputation so that the accuracy of imputation can be improved. Unlike KEMI and KEMI[+], they divide the entire dataset into complete and incomplete records and then impute the missing values using only complete records, so if the dataset does not contain any complete record, the algorithms will stop and will not be able to impute any incomplete record. DMI and KDMI also divide the entire dataset into complete and incomplete records and then build trees from complete records only, so if the dataset does not contain any complete record, the algorithms will not be able to build any tree. They will not be able to impute any incomplete records. The proposed imputation techniques, KI and FCKI, can only handle numerical features, not categorical features.

FCKI requires less computational time than other advanced missing data imputation methods; DMI, KDMI, KEMI, KEMI+, KI because FCKI searches a cluster, instead of the whole dataset, to find the best k-nearest neighbors. The average execution time in FCKI is better than DMI by 34.11%, KDMI by 70.46%, KEMI by 86.23%, KEMI[+] by 88.6%, and KI by 86.72%. DMI and kDMI make utilization of a decision tree for the horizontal partitioning. This is computationally expensive. They build a decision tree for each feature having missing values in the dataset, considering this feature as a class attribute. If there are a large number of features having missing values, DMI and KDMI will build a large number of trees, even if the dataset contains a small number of records.

Table 10 shows a comparison between KI and FCKI. The reported results in Table 10 show that the average RMSE value overall datasets of FCKI is less than KI because FCKI applies two levels of similarity before imputing the missing values while KI applies only one level of similarity. So FCKI can achieve a higher imputation accuracy. The reported results in Table 10 show also that FCKI has lower average execution time

**Table 10  A comparison between KI and FCKI.**

| Proposed imputation method | KI | FCKI |
|---|---|---|
| Purpose | Improving missing data imputation accuracy only. | Improving time efficiency as well as imputation accuracy. |
| Components | • It integrates k-nearest neighbors and iterative imputation.<br>• It applies only one level of similarity using kNN algorithm. | • It integrates fuzzy c-means, k-nearest neighbors, and iterative imputation.<br>• It applies two levels of similarity using FCM and kNN algorithms. |
| Average RMSE value | 0.0567 | 0.0521 |
| Average execution time (S) | 77743 | 10323 |

overall datasets than KI. because FCKI searches a cluster, instead of the whole dataset, to find the best k-nearest neighbors while KI is expensive because it searches within the entire dataset to find the most similar records for each missing record.

## CONCLUSIONS AND FUTURE WORK

Two missing data imputation methods are proposed in this work. The first technique, called KI, consolidates k-nearest neighbors and iterative imputation algorithms for imputing the missing data for a dataset. This technique applies only one level of similarity using the kNN algorithm. A suitable k value is estimated automatically for the kNN. The best k-nearest neighbors for each missing record is discovered based on records similarity. The iterative imputation method is then used to impute the missing values of the incomplete records by using the global correlation structure among the selected records. This technique can improve missing data imputation accuracy. However, it is expensive for a large dataset because it is required to search within the entire dataset to find the most similar records for each missing record, so an enhanced hybrid missing data imputation method is proposed, called FCKI, which is an extension of KI. It integrates fuzzy c-means, k-nearest neighbors, and iterative imputation algorithms for imputing the missing data for a dataset. This technique focuses on improving time efficiency for the proposed missing data imputation algorithm as well as missing data imputation accuracy. It uses fuzzy c-means clustering for the dataset to divide records of the dataset into c fuzzy clusters where the records in the same cluster are more similar to each other. Then, it imputes each cluster separately using the KI algorithm. FCKI applies two levels of similarity. This technique has the advantage of tackling missing values based on the similarity of the set of records instead of the whole dataset. FCKI can improve time efficiency because It does not have many iterations for imputing missing values in the dataset. It also focuses on both the similarity of data records and the correlation among the features. Therefore, it can enhance imputation efficiency and effectiveness where the most efficient imputation method should impute incomplete dataset with the least amount of time and the most effective imputation method should achieve the highest imputation accuracy.

The performance of the proposed imputation methods is assessed by experimenting them with fifteen available datasets through various missing ratios for each type of missing data; MCAR, MAR, MNAR, and, then, compared with ten competitors in terms of three imputation performance measures, which are the root mean square error (RMSE), the normalized root mean square error (NRMSE), and the mean absolute error (MAE). The attained results show that the proposed imputation techniques, KI and FCKI, outperform other competitors in terms of imputation accuracy. The average RMSE values in KI and FCKI are better than KEMI[+,] which is the best competitor by 35.42% and 40.69%, respectively. The results also show that FCKI requires less computational time than other advanced missing data imputation methods; DMI, KDMI, KEMI, KEMI[+], KI. The average execution time in FCKI is better than DMI by 34.11%, KDMI by 70.46%, KEMI by 86.23%, KEMI[+] by 88.6%, and KI by 86.72%.

The proposed imputation techniques, KI and FCKI, can only handle numerical features. The proposed schemes could also be expanded to use imputation methods that can deal with categorical features and heterogeneous datasets to impute different features, which appear to be useful directions for future research. Future work also includes evaluating the proposed methods on real big datasets to compare the accuracy on varieties of datasets. The behavior of the proposed imputation techniques needs to be evaluated for missing ratios greater than 20% in order to see if this can affect the quality of the missing data imputation. We also plan to explore whether the proposed imputation techniques, KI and FCKI, is useful for data mining tasks such as classification. In order to reduce the computational time, the imputation process for the FCKI algorithm can be parallelized and we will also try to replace kNN with another method such as support vector machine (SVM) or random forests (RFs) to reduce the computational time for imputing the missing values in a large-scale dataset while maintaining the high imputation accuracy.

### Funding
The authors received no funding for this work.

### Competing Interests
The authors declare that they have no competing interests.

### Author Contributions
- Khaled M. Fouad conceived and designed the experiments, performed the experiments, analyzed the data, performed the computation work, prepared figures and/or tables, authored or reviewed drafts of the paper, and approved the final draft.
- Mahmoud M. Ismail conceived and designed the experiments, performed the experiments, analyzed the data, performed the computation work, prepared figures and/or tables, authored or reviewed drafts of the paper, and approved the final draft.

- Ahmad Taher Azar conceived and designed the experiments, analyzed the data, prepared figures and/or tables, authored or reviewed drafts of the paper, and approved the final draft.
- Mona M. Arafa conceived and designed the experiments, analyzed the data, prepared figures and/or tables, and approved the final draft.

## Data Availability

The datasets and the Python source code files for generating different missing data types (MCAR, MAR, MNAR) and for missing data imputation methods used in the experiments are available in the Supplemental Files.

## Supplemental Information

Supplemental information for this article can be found online at http://dx.doi.org/10.7717/peerj-cs.619#supplemental-information.

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
