# Peer review of "Advanced methods for missing values imputation based on similarity learning"

_PeerJ Computer Science, doi:10.7717/peerj-cs.619_

## Round 0.1 · original submission · Major Revisions

According to the reviewers, the paper has its merits, but there are several points that should be improved such as (i) problem definition, (ii) related work, and (iii) experiments. I also suggest that the authors get editing help from someone with full professional proficiency in English.

Reviewer 1 ·

Basic reporting

- In the abstract, please add the stands for all of the abbreviations.

- The conclusion should present summarize instead of repeating the detail of the proposed approach

Experimental design

Please specify what is parameters of the dataset should be categorized as large-scale and small datasets, also there should provide a comparison of the difference if the method is used in the small and large data sets.

Validity of the findings

- It is not clear the correlation between KI and FCKI, please consider adding the general diagram that describes the proposed architecture in brief.

- Please provide an ablation study to explain the details for each component in the proposed approach

Additional comments

The problem is clearly defined in this paper with two proposed approached called KI and FCKI. Note from me related to the proposed approach is the correlation between KI and FCKI should be present more clearly.

Reviewer 2 ·

Basic reporting

This paper describes two similarity-based methods to impute missing data. The text is well structured and reports promising results. Yet, some issues must be pointed out:


General comments

- Text needs proofreading. There are some minor grammatical mistakes, typos and incomplete sentences. Please check the comments in the document for examples of such problems.

- There are several sentences that should be supported by references. There are highlighted examples of these sentences in the document.
- Section 3 also encompasses related work. Why is it apart from Section 2?

- Placing figures and tables at the end of the manuscript severely hinders the reviewing process. It demands many ups and downs. Please, put them near their citations.

- Numbering the steps described in Figures 1 and 2 would help the reader a lot.

- The text references appendices and tables that are not available. Please, check the comments about these flaws in the document.



Content/technical comments

Abstract

- I think that there is one important gap in the abstract (and in the rest of the paper):
The text does not state the problem clearly. There are several studies about missing data imputation that are based on data similarity. What are the limitations of those studies that the methods proposed by this paper overcome? The abstract (and the rest of the paper) must highlight it clearly. I believe line 644 may contain a glance of what must be clarified.
- In accordance with the previous comment, the statement “The primary existing techniques provide low efficiency and effectiveness results for imputing an incomplete large-scale dataset.” should be better justified.

Section 1

- Line 48: A dataset is not trained.
- Line 100 (and several others): I think the word "phase" is not appropriate, once it suggests some kind of temporal ordering which does not seem to be the case. I believe the proposed methods can be applied independently.
- KI seems to follow a classical hot-deck approach to impute data based on data similarity (by the way, this approach was never mentioned in any part of the text – it indicates a several flaw in the literature reviewing process). How does KI differ from KNNI, for example? What is the hypothesis (ie., the rationale) that makes the authors believe that this method can overcome the state-of-the-art ones? This is a very important point and must be clarified.

- The same question holds for FCKI.
- Another question concerning FCKI: If the dataset is big, the clustering process may have high computational demand. This point must be better justified.
- Do the proposed methods consider datasets with missing values in multiple attributes? Please, clearly state that.



Sections 2 and 3

- These sections do not clearly state the limitation(s) of the existing imputation methods that is (are) tackled by the proposed methods.
- In my opinion, related work should be presented and analyzed under the problem’s point of view. Please, explicitly (and exclusively) mention related work’s limitations concerning the problem’s point of view. Limitations that are not treated by the article should not be mentioned.
- What about hot-deck-based data imputation methods?
- Here go other studies about missing data imputation methods. They may be included as related work:

* Christopher, S. Z., Siswantining, T., Sarwinda, D., and Bustaman, A. (2019). Missing value analysis of numerical data using fractional hot-deck imputation. In 2019 3rd International Conference on Informatics and Computational Sciences (ICICoS), pages 1–6. IEEE.

* Ford, B. L. (1983). An overview of hot-deck procedures. Incomplete data in sample
surveys, 2(Part IV):185–207.
* Graham, J. W. (2012). Missing data: Analysis and design. Springer Science & Business
Media.

* Graham, J. W., Cumsille, P. E., and Shevock, A. E. (2012). Methods for handling missing
data. Handbook of Psychology, Second Edition, 2.
* Hassan, M. M., Atiya, A. F., El-Gayar, N., and El-Fouly, R. (2007). Regression in the
presence missing data using ensemble methods. In 2007 International Joint Conference
on Neural Networks, pages 1261–1265. IEEE.

* Karanikola, A. and Kotsiantis, S. (2019). A hybrid method for missing value imputation. In
Proceedings of the 23rd Pan-Hellenic Conference on Informatics, pages 74–79.
* Kumar, S., Pandey, M. K., Nath, A., and Subbiah, K. (2016). Performance analysis of
ensemble supervised machine learning algorithms for missing value imputation. In
2016 2nd International Conference on Computational Intelligence and Networks (CINE),
pages 160–165. IEEE.
* Lu, X., Si, J., Pan, L., and Zhao, Y. (2011). Imputation of missing data using ensemble
algorithms. In 2011 Eighth International Conference on Fuzzy Systems and Knowledge
Discovery (FSKD), volume 2, pages 1312–1315. IEEE.

* Madhu, G., Bharadwaj, B. L., Nagachandrika, G., and Vardhan, K. S. (2019). A novel
algorithm for missing data imputation on machine learning. In 2019 International
Conference on Smart Systems and Inventive Technology (ICSSIT), pages 173–177.
IEEE.

* Magnani, M. (2004). Techniques for dealing with missing data in knowledge discovery
tasks. Obtido http://magnanim. web. cs. unibo. it/index. html, 15(01):2007.
* Poulos, J. and Valle, R. (2018). Missing data imputation for supervised learning. Applied
Artificial Intelligence, 32(2):186–196.
* Rubin, D. B. (1988). An overview of multiple imputation. In Proceedings of the survey
research methods section of the American statistical association, pages 79–84. Citeseer.
* Suresh, M., Taib, R., Zhao, Y., and Jin, W. (2019). Sharpening the blade: Missing data
imputation using supervised machine learning. In Australasian Joint Conference on
Artificial Intelligence, pages 215–227. Springer.

* Templ, M., Kowarik, A., and Filzmoser, P. (2011). Iterative stepwise regression imputation
using standard and robust methods. Computational Statistics & Data Analysis,
55(10):2793–2806.

Experimental design

Section 5
- Line 322: missing record or missing value?
- Suggestion: Present the definitions in a separate table before the explanations given in lines 341- 367. It would help a lot.
- Be careful with some formalisms. E.g.:
* Is R_i the set of records returned by KNN for record r_i? Or R_i=r_i? There seems to be a notation flaw here.
* What is a k-NN record?
* kNN=k-NN?
* Line 347: Consider removing "{" and "}". Moreover, R_i is a record, not a set... This problem occurs in other points. See highlighted comments in the document.
- Line 357 - Suggestion: Use "N_p - 1 RMSE values" instead of "values of N_p - 1 RMSE"
- Is there a loop to consider multiple non-missing value attributes in R_i? What about step 11 in this case?
- Algorithm 4: What happens if there are many values for z?
- Considering FCKI: What happens when a data record belongs to two or more clusters? According to the fuzzy set theory, it is possible and, hence, must be taken into consideration.
- Line 414: Does it consider the membership degree of each record to the corresponding cluster?

Validity of the findings

Section 6
- Table 1: None of these datasets actually contains "big" data. 100000 records (in PPS - the biggest dataset) cannot be considered a high value when compared to most contemporary real-world scenarios. Hence, experiments with real big datasets should be executed.
- Table 1: Although the values of the features in the Zoo dataset have numerical representation, they are, in fact, boolean (i.e., categorical) values. This should be better explained. Moreover, the other datasets must be checked concerning this subtle characteristic.
- Line 498: Why sonar?
- Table 2: What about the standard deviation?
- Table 2: Please, highlight the best value for each dataset.
- What about a detailed analysis of the boxplot of Figure 9?
- Is there a reason that justifies KI and FCKI's superiority?
- What about a comparison between KI and FCKI?
- The authors should comment on NRMSE and MAE values.
- I suggest that the authors run the Friedman Test before presenting the results of the Nemenyi Test. By the way, what of the confidence value used in the test? What about the values of the statistic?
- Lines 635-639: Are those future works? If so, I suggest placing them at the end of Section 7.
- Table 5: Does this time encompass finding the best c value and the clustering process? Please clarify that in the text.

Annotated reviews are not available for download in order to protect the identity of reviewers who chose to remain anonymous.

·

Basic reporting

-> Clear and unambiguous, professional English used throughout.
Generally, the article uses clear, adequate technical language. It respects the style of scientific communications and has an easy reading.

However, the excessive use of the verb "to tackle", especially in the first part of the text, makes reading truncated. Synonyms can quickly solve this situation.

There is also an overuse of passive voice in the text, which can be tiring for the reader. More direct communication becomes useful in its purposes. Finally, there is an excessive repetition of words. Make more use of synonyms.

-> Literature references, sufficient field background/context provided.
The core of the proposal omitted a vital concept: both proposed implement hot-deck imputation, proposed by Ford in 1983 (Ford, BL. An overview of hot-deck procedures. In: Madow, WG .; Olkin, I .; Rubin, DB., Editors. Incomplete Data in Sample Surveys. Vol. 2. New York: Academic Press; 1983. p. 185-207).

The authors' proposals reveal that they adopt the hot-deck imputation. It becomes apparent in the abstract, in lines 19 to 21: "The best set of nearest neighbors for each missing record is discovered through the records similarity by using the KNN algorithm."

The work uses the k-NN and fuzzy c-means algorithms as predecessors of imputation. However, they are clustering algorithms! So, we see the hot-deck imputation there.

It is the most severe structural failure that the article presents since it ignores 37 years of research that used the term hot-deck to reference algorithms that group related data, reducing the original data set before inputting them to provide better accuracy in filling in missing data.

From this demand, there is a critical need to review the literature, including hot-deck approaches. It affects the entire literature review decisively.

The entire article should be revised based on this concept. Several papers approach this paradigm.

-> Professional article structure, figures, tables. Raw data shared.
The structure of the article follows the standard expected for scientific publications. The sections have, in general, a well-defined function in the context of the work. Figures and tables contribute to the understanding of the proposal and the analysis of results.

However, Section 3 ("Imputation Methods") seems out of context when reading the article sequentially. After completing the work's reading, its function presents characteristics of the ten methods chosen for comparison with the authors' proposal. However, this looks more like a hybrid section of theoretical foundation with the works that proposed the methods that serve as the analysis target. The name "Imputation Methods" does not reveal the semantics of the section.

Another significant observation is related to the quality of figures 1 and 2, the authors' proposals heart. Understandably, there is a spatial challenge to be faced (the page size). However, given the images' complexity, understanding the proposal just looking for the figures was a challenging task. Perhaps the authors should divide the two figures of the proposals into functional blocks and adding new figures. The pollution of current visual and textual elements will not make the article easy to read for the future audience.

Besides, the images' physical disposition, far from the text's natural flow, greatly affected this analysis. The comings and goings between the text that detailed the algorithm and the figures, coupled with the overload of each figure's visual elements, made it very difficult to assess.

-> Self-contained with relevant results to hypotheses.
The presented proposal can be considered self-contained. It contextualizes the problem, defines the terms, algorithms, and metrics it uses. However, the text does not present the hypothesis, the addressed research problem. We start with the proposal of two new imputation presented methods as more efficient. However, the problem itself is not formally explained.

-> Formal results should include clear definitions of all terms and theorems and detailed proofs.
The results presented use concepts and definitions that were previously formally defined, and the algorithms used to generate missing data.

Experimental design

-> Original primary research within Aims and Scope of the journal.
The article serves the journal's objectives and scopes well. It can be categorized correctly under the theme "Data Mining and Machine Learning." Regarding originality, the article makes an alternative proposal for data imputation using two new methods. After a previous search, these methods were, in fact, not previously proposed in the manner presented, which guarantees the originality of the work.

-> Research question well defined, relevant & meaningful. It is stated how research fills an identified knowledge gap.
The research question was not clearly defined. This inference comes from the proposal, but there is no clear presentation of a hypothesis – a situation that is easy to solve but needs attention.

-> Rigorous investigation performed to a high technical & ethical standard.
The work has a satisfactory performance in technical and ethical rigors, without showing signs of disappointment in this regard.

-> Methods described with sufficient detail & information to replicate.
The influential contributions of the proposal revolve around the choice of the best k value in the first proposed method (KI), and the second in the use of fuzzy c-means in the FCKI method, which allows the same occurrence (tuple) to participate in several groups of the later phase for the use of k-NN.

The first contribution appears between lines 665 and 671 ("By comparing with the original k-means for imputing the missing data, there are some benefits for fuzzy clustering. Fuzzy clustering is more naturalistic than hard clustering in many situations. When records are not well-separated, it provides a better description method, as is the case for missing data problems. Besides, suppose the initial points are not correctly selected. In that case, the original k-means algorithm may be stuck in the minimum local status. Continuous membership values in the fuzzy clustering, on the other hand, provide the resulting algorithms less sensitive to get stuck at a local minimum"). This conclusion comes from the analysis of Figure 1, from this highlighted section, and from line 644 ("A suitable k value is estimated automatically for the k-NN"). I had to deduce that after several rounds of reading. We need to improve this presentation.

The second contribution was evident in the section between lines 385 and 389 ("In the case of fuzzy clustering techniques, records on the frontiers between multiple clusters are not compelled to assign to one of the clusters completely. The records can belong to multiple clusters at the same time (Sefidian & Daneshpour, 2019). Each record has a membership degree between 0 and 1, showing its partial membership"). It seems that the ideas could be better presented in the article. It should be a consequence of the lack of definition of a research question, a hypothesis. The proposed methods present a set of questionable choices. Some highlighted points:

- Why did the absence percentages vary between 1% and 20%? Is there a significant difference in the method's performance between 1% and 2%, or between 19% and 20%? Moreover, do the proposed methods work well with higher percentages of absence, where the base's semantic loss is accentuated (which can influence the imputed data quality)?

- The authors chose ten methods for comparison (mean, kNNI, SoftImpute, SVDimpute, Iterative Imputation, EMI, DMI, KDMI, KEMI, and KEMI+), but did not adequately justify this choice. Between lines 129 and 135, the methods are presented and referenced, but there is no mention of the selection criteria. Between lines 269 and 271, the authors reinforce that the chosen methods as competitors. The impression that remains is that the choice of comparison algorithms left out a more straightforward analysis – especially when it comes to performance evaluation. Why didn't they compare KI and FCKI with hot-deck imputation using a clustering algorithm like k-means, for example?

- the article generates an expectation of discussion regarding computational time, which proved to be reasonably deflated. In lines 576 and 577, they affirm that "The experiments investigate the scalability of the missing data imputation methods by assessing runtimes for all datasets with different missing ratios [...]". However, what kind of time is measured? Wall time? Processor time? When discussing performance, there is a need at least to refer to speedup issues, which did not appear. Given that the authors pointed the discussion in this direction, it deserves more outstanding care.

- Presenting missing value generation algorithms MCAR, MAR, and NMAR is exciting. Most papers that simulate them do not present the way it works. So, the authors' initiative, who based on the work of Garciarena and Santana (2017), referenced in the text, should be praised. However, some issues need clarification regarding this point: (i) I was unable to identify the adaptation mentioned (line 286: "These algorithms are derived from Garciarena & Santana (2017)"). It seems that it is only a detailed form of them, presented in the form of pseudocode. Thus, there are two paths to follow: either the authors present the adaptations made to the algorithm that makes a difference or mention them and initially present them. Garciarena and Santana's article shows the algorithms in a cleaner, intelligible form. The one presented by the authors may appear in an appendix, but not in the main text, under pain of tiring the reader; (ii) on lines 298 and 299, the authors say that "The algorithm presumes a single attribute as causative." As presented, it seems to be a proposal from them, the authors, and not from Garciarena and Santana's paper. So, it is necessary to complement the information; (iii) in the MNAR algorithm (Algorithm 3), the authors ignore the discussion made by Garciarena and Santana, who share the absence mechanism in MIV and MuOV. Algorithm 3 implements which of the two sub-options?

- lines 29 and 32: "Therefore, proposed imputation techniques are compared with other missing data imputation methods using three measures; root mean square error (RMSE), the normalized root mean square error (NRMSE), and the mean absolute error (MAE)." This statement was mentioned six more times in the text (lines 136 to 138, lines 272 to 274, lines 487 to 489, lines 521 to 523, lines 550 to 552, and lines 675 and 676). Despite so many mentions, the whole discussion of accuracy revolved around RMSE, the preferred metric for imputation articles. What, after all, why calculate NRMSE and MAE metrics? The only use I was able to notice was in the generation of boxplots

- on lines 664 and 665, the authors say: "Therefore, it can enhance imputation effectiveness and efficiency." What differentiates the two concepts in the article? It certainly is a critical discussion.

- on line 567, the authors indicate the use of the Nemenyi non-parametric test. Why didn't they choose the Friedman test, which is also non-parametric and more popular? Nemenyi is a test by nature conservative. There is at least an explanation of the reason for the choice.

Validity of the findings

-> Impact and novelty not assessed. Negative/inconclusive results accepted. Meaningful replication encouraged where rationale & benefit to literature is clearly stated.
The reproduction of the experiments follows the pattern used in the research field on missing data. Furthermore, considering that there is no benchmark to reference, which would mention previously known results of baseline algorithms' execution, each experiment's re-execution is necessary.

-> All underlying data have been provided; they are robust, statistically sound, & controlled.
The authors attached the databases they used in the article. Fifteen files in the CSV format were available on the downloadable platform and the Python code that implements the missing data generator algorithms. Every file has no flaws. Table 1 offers the attached databases' essential characteristics, except for the files' size. It is also essential data to figure out in the table. This information (size of each dataset) could appear as a fourth column instead of the "Number of categorical features" column, equal to zero for all cases.

The most extensive dataset is 25 MB; the one with the most records, 100,000 tuples; the one with the most columns has 90 features.

I missed a more consistent justification regarding the choice of these bases. In subsection 6.2, presenting the selection criteria for the bases lines 471 and 472 only indicate "The evaluation of the proposed methods to impute missing data, KI and FCKI, using fifteen datasets, shown in Table 1". Why were these bases chosen? What repository did they come from? Is there any basis that justifies that they fit well with the objectives of what they want to demonstrate? Were they used in previous works that use the ten algorithms chosen to compare? The only explanation was found on line 479 was in the excerpt "The selected databases are commonly used." Commonly under what conditions? Where are the references of the communal use of these datasets? It is not a real justification. The absence of a methodological approach to choosing the bases may generate sampling bias.

In the excerpt "The attained results show that the proposed imputation techniques, KI and FCKI, outperform other competitors in terms of imputation accuracy," a more consistent discussion on the choice of bases would give more significant support to this statement.

The study of imputation techniques on small datasets is essential and does not invalidate the results obtained. Besides, the use of small databases is a widespread feature in imputation works. However, it must not need a more careful analysis not to jeopardize the research's contributions. It is a challenge that the work in the area needs to solve in general.

Furthermore, there was an expectation that the databases used were indeed large, given the excerpt "In the analysis and mining of large-scale data, large missing values present formidable challenges" presented in lines 45, which was confirmed neither in the presentation of the work nor in the datasets. It was evident looking for the computing environment was described in section 6 – a machine with a 2x4 core Intel i7-7500U processor with 8 GB of RAM. These are characteristics of personal computer desktops or notebooks, revealing the lack of demand for greater computational power equipment.

This expectation matches the text between lines 110 and 113: "In the second phase, an enhanced hybrid missing data imputation method is proposed, called FCKI, an extension of KI. It integrates fuzzy c-means, k-nearest neighbors, and iterative imputation method for imputing the missing data for a large-scale dataset".

The visualization of datasets in sonar charts is an exciting differential of the article, not seen in previous works. It gives a good view of the patterns of absence, restricted, however, pinning in an absence percentage (10%). Nevertheless, why were this value chosen? How it works on others missing values percentages? How is the generation of these graphs reproduced?

-> Conclusions are well stated, linked to original research question & limited to supporting results.
Conclusions follow what is expected: they review the motivations, consolidate the results, and point to future work.

The authors were shy when presenting future works. There are several possibilities to explore.

-> Speculation is welcome but should be identified as such.
There are some strong speculations not identified as such in the article:

1. In lines 13 and 14, the authors state that "The main challenge of mining datasets is the existence14 of missing values". This is a strong statement that needs to be referenced or proved.

2. In lines 15 and 16, the statement "The primary existing techniques provide low efficiency and effectiveness results for imputing an incomplete large-scale dataset" also needs to be supported, as it disappoints an entire history of related works in the area.

Additional comments

1. I did not understand the excerpt "A method of missing data imputation is considered efficient. It means that it does not have to rely on the entire dataset for imputing missing records" between lines 97 and 99. It seems lost in the paragraph.

2. Describing the same algorithm twice is tiring for the reader and generates an increase in the article's size that is not interesting. It happens with algorithms 4 and 5.

3. In this same section, step 1 mentions the use of the elbow method. Why is it not described and referenced? Is it a prerequisite for the reader to know?

4. The excerpt "The missing values are generated into the selected datasets using various algorithms, which generate various missing patterns" in lines 480 and 481 induces a deconstruction presented in Section 4. I believe the whole job considered the algorithms presented. So, it is essential always to emphasize them.

5. The sentence between lines 490 and 491 should appear in the previous paragraph.

6. The paragraph between lines 633 and 639 is misplaced. The first part (justifying k-NN) should appear in the proposal section and the second part in future works.

7. The term "global correlation" used in line 647 conflicts with the idea of global imputation enshrined in the literature. This considers the use of the entire dataset suggests reviewing the term's use.

---

## Round 0.2 · Minor Revisions

As you are aware, your article received a "Minor Revisions" status. Nevertheless, it is fundamental to consider the comments of the reviewers in the new version to be submitted.

Reviewer 1 ·

Basic reporting

- The author has already explained the comment on the first revision. Please check again and make sure if the writing style of the algorithm name is consistent, e.g., "k nearest neighbor imputation" should written as "k-nearest neighbor," and please consider that "k" is the name of a variable.

- The author has already modified the conclusion that summarizes the proposed approach and its results.

Experimental design

- "Large-scale data can be associated with data that grows to a huge size over time. Small data is data in a volume and format that makes it accessible, informative, and actionable." The quoted statement is obtained from the rebuttal letter should be mentioned in the manuscript to clarify the different small and large scale datasets.

Validity of the findings

the author already re-create the figure to make a clearer representation of the correlation of FI and FCKI

Additional comments

In general, the response from the author is addressing the reviewer's question. In some points, the author should be considered some advice to be reviewed for the better manuscript.

·

Basic reporting

All the answers presented fully comply with the points previously raised.

Experimental design

- Why did the absence percentages vary between 1% and 20%? Is there a significant difference in the method's performance between 1% and 2%, or between 19% and 20%? Moreover, do the proposed methods work well with higher percentages of absence, where the base's semantic loss is accentuated (which can influence the imputed data quality)?

Response: As shown in the results, the performance varies through different missing ratios from 1% to 20% for the three types of missing data; MCAR, MAR, MNAR. These results show the proposed method's performance by applying twelve different datasets generated by using three types of missing data; MCAR, MAR, MNAR for four different missing ratios; 1%, 5%, 10%, and 20%. The missing data imputation review shows that the familiar and significantly different missing ratios used to apply the imputation methods are from 1% to 20%.

Rejoinder: It is understandable the author's choices. Nevertheless, it is essential that at least one comment touch this question fits in future works. The algorithm's behavior needs to be evaluated for data absence rates greater than 20% since the semantic loss of the dataset can affect the results obtained.


- The authors chose ten methods for comparison (mean, kNNI, SoftImpute, SVDimpute, Iterative Imputation, EMI, DMI, KDMI, KEMI, and KEMI+), but did not adequately justify this choice. Between lines 129 and 135, the methods are presented and referenced, but there is no mention of the selection criteria. Between lines 269 and 271, the authors reinforce that the chosen methods as competitors. The impression that remains is that the choice of comparison algorithms left out a more straightforward analysis – especially when it comes to performance evaluation. Why didn't they compare KI and FCKI with hot-deck imputation using a clustering algorithm like k-means, for example?

Response: The hot-deck imputation involves replacing missing values with values from a "similar" responding unit. Therefore, KNNI and KI demonstrate an essential hot-deck imputation to tackle missing data based on data similarity. We think that hot-deck imputation is found implicitly in our comparison. We have also included other hot deck-based imputation methods in the related work section.

Rejoinder: The highlighted question specifically touches less on the question more about the justification for using the ten mentioned algorithms. It is comprehensible that it is essential to explain why this work had chosen each of these algorithms and not others. A well-founded explanation strengthens the work presented, providing guidance for future research in the area.

Validity of the findings

- The study of imputation techniques on small datasets is essential and does not invalidate the results obtained. Besides, the use of small databases is a widespread feature in imputation works. However, it must not need a more careful analysis not to jeopardize the research's contributions. It is a challenge that the work in the area needs to solve in general.
Furthermore, there was an expectation that the databases used were indeed large, given the excerpt "In the analysis and mining of large-scale data, large missing values present formidable challenges" presented in lines 45, which was confirmed neither in the presentation of the work nor in the datasets. It was evident that looking for the computing environment was described in section 6 – a machine with a 2x4 core Intel i7-7500U processor with 8 GB of RAM. These are characteristics of personal computer desktops or notebooks, revealing the lack of demand for greater computational power equipment.
This expectation matches the text between lines 110 and 113: "In the second phase, an enhanced hybrid missing data imputation method is proposed, called FCKI, an extension of KI. It integrates fuzzy c-means, k-nearest neighbors, and iterative imputation method for imputing the missing data for a large-scale dataset".

Response: Concerning the first observation, we used the algorithm to validate the model with different dataset sizes.
We fully agree that the application on large real datasets is helpful to compare the exactness of different data sets. We collected some real datasets, but we could not access them at our laboratories due to the COVID-19 lockdown. We hope to understand the critical circumstances worldwide and how most laboratories are locked because of social distance restrictions.
We will take your valuable comment into consideration as a future direction to extend our work on actual big data. In the section on Conclusions and Future Work, we also mentioned that point.
Concerning the second observation, a 2x4 core Intel Processor I7-7500U with 8 GB RAM was used. We agree entirely that the computer should be used better for quick computing on other high-performance computers. At Benha University, we already have a fully equipped computer intelligence lab with high-performance computers and fast GPUs. However, due to COVID-19 situations and limited government rules, we could not use the laboratory.
We will look at your valuable point for future work and hope that the situation will improve in the next period to achieve all our goals.

Rejoinder: I really understand the exceptional situation that we are experiencing. It imposes difficulties that we would not usually have. However, the text still gives the impression that it claims to work with large datasets when that is not the situation at hand. Thus, I believe that it is interesting that the authors reevaluate the use of the term "large" together with datasets to resolve doubts such as those I present.

Additional comments

All the answers presented fully comply with the points previously raised.

---

## Round 0.3 · accepted · Accept

According to the reviewers, all modifications in the paper fully comply with the points previously raised. This way, this paper can be accepted.

Please consider the final comments from Reviewer 3.

Reviewer 1 ·

Basic reporting

no comment

Experimental design

no comment

Validity of the findings

no comment

·

Basic reporting

All the answers presented fully comply with the points previously raised.

Experimental design

- The authors chose ten methods for comparison (mean, kNNI, SoftImpute, SVDimpute, Iterative Imputation, EMI, DMI, KDMI, KEMI, and KEMI+), but did not adequately justify this choice. Between lines 129 and 135, the methods are presented and referenced, but there is no mention of the selection criteria. Between lines 269 and 271, the authors reinforce that the chosen methods as competitors. The impression that remains is that the choice of comparison algorithms left out a more straightforward analysis – especially when it comes to performance evaluation. Why didn't they compare KI and FCKI with hot-deck imputation using a clustering algorithm like k-means, for example?

Response: The hot-deck imputation involves replacing missing values with values from a "similar" responding unit. Therefore, KNNI and KI demonstrate an essential hot-deck imputation to tackle missing data based on data similarity. We think that hot-deck imputation is found implicitly in our comparison. We have also included other hot deck-based imputation methods in the related work section.

Rejoinder: The highlighted question specifically touches less on the question more about the justification for using the ten mentioned algorithms. It is comprehensible that it is essential to explain why this work had chosen each of these algorithms and not others. A well-founded explanation strengthens the work presented, providing guidance for future research in the area.
Response: Thanks for your valuable comment. We used the existing methods because the literature review mentioned that there is a need to be applied for many applications with different sizes of datasets and they have been shown to be effective. Some recent research relied on these methods for comparison. Moreover, most of these methods rely on data similarity, which is the basis of our proposed methods. These methods gradually vary from simple to advanced. In this revised version, we have clarified this point in lines 555-560.

Rejoinder: To complete this explanation, it is important that the authors indicate articles that implement these techniques after the sentence "Some recent research relied on these methods for comparison" (lines 557-558). These references must touch the listed algorithms.

Validity of the findings

All the answers presented fully comply with the points previously raised.

Additional comments

All the answers presented fully comply with the points previously raised.